# Dynamics of DNA damage-induced nuclear inclusions are regulated by SUMOylation of Btn2

Arun Kumar[1,2], Veena Mathew[1] & Peter C. Stirling [1,2] ✉

Spatial compartmentalization is a key facet of protein quality control that serves to store disassembled or non-native proteins until triage to the refolding or degradation machinery can occur in a regulated manner. Yeast cells sequester nuclear proteins at intranuclear quality control bodies (INQ) in response to various stresses, although the regulation of this process remains poorly understood. Here we reveal the SUMO modification of the small heat shock protein Btn2 under DNA damage and place Btn2 SUMOylation in a pathway promoting protein clearance from INQ structures. Along with other chaperones, and degradation machinery, Btn2-SUMO promotes INQ clearance from cells recovering from genotoxic stress. These data link small heat shock protein post-translational modification to the regulation of protein sequestration in the yeast nucleus.

Maintaining protein homeostasis (proteostasis) is a fundamental property of all cells that is essential for survival. Exposure to stressors such as DNA damage, heat, mutations, or aging can cause proteins to acquire non-native conformations. These misfolded proteins feature hydrophobic stretches that can accumulate to form toxic protein aggregates that eventually disrupt various molecular pathways. This interference in cellular function is particularly evident in neurodegenerative diseases such as Alzheimer's, amyotrophic lateral sclerosis, and Parkinson's, which are linked to protein aggregation pathology[1–5]. Therefore, cells have evolved to maintain proteostasis through a network of protein quality control (PQC) circuits that employ molecular chaperones to either refold, degrade, or sequester misfolded proteins.

It was thought that PQC relies only on protein refolding, degradation, and autophagy to maintain a flux of amino acids and proteostasis. This notion has since been challenged since the discovery of protein sequestration[6,7]. Under severe stress conditions that overburden PQC systems, misfolded proteins are spatially sequestered to specialized and distinct membrane-less inclusions within the cell. This controlled aggregation process includes relocalization of chaperones involved in refolding and degradation pathways suggesting that sequestration could help improve local concentration of PQC factors to promote stress recovery. In *Saccharomyces cerevisiae*, misfolded proteins have been found to partition into one of three inclusions:

Nuclear misfolded proteins are sequestered to the peri-nucleolar intranuclear quality control site (INQ), whereas cytoplasmic proteins can be sequestered to two distinct inclusions based on their solubility. If soluble, cytoplasmic misfolded proteins are ubiquitinated and sequestered to the peri-nuclear juxtanuclear quality control site (JUNQ) while insoluble proteins are targeted to the peri-vacuolar insoluble protein deposit (IPOD)[8,9]. Multiple small cytoplasmic foci containing misfolded proteins have also been reported, but are thought to coalesce to form the JUNQ[10]. While these inclusions are site-specific, there have been reports of interplay such that cytoplasmic misfolded proteins can be shuttled between the JUNQ to INQ[7,11]. Furthermore, the INQ and JUNQ have been found to home to the nuclear-vacuolar junction supporting this idea that there can be transfer of misfolded proteins to and from different sequestration sites[11].

The spatiotemporal organization of misfolded proteins is orchestrated by small-heat shock proteins (sHsps) Btn2 and Hsp42 in *S. cerevisiae*[7,12–14]. Studies utilizing terminally misfolded proteins (e.g., VHL, Ubc9-tsGFP) have uncovered site-specific roles of these sHsps such that Hsp42 targets proteins to cytoplasmic inclusions whereas Btn2 sequesters proteins to INQ. Interestingly, endogenous proteins have also been found to relocalize to INQ under replication stress and heat shock coupled with proteasomal disruption[15–19]. While only a small number of proteins have been localized to INQ using cytological

[1]Terry Fox Laboratory, BC Cancer, 675 West 10th Avenue, Vancouver, BC V5Z1L3, Canada. [2]Department of Medical Genetics, University of British Columbia, Vancouver, BC V6T1Z4, Canada. ✉e-mail: pstirling@bccrc.ca

screening of GFP fusions, these studies have helped define factors involved in INQ PQC, namely the competition between Apj1-Hsp70 assisted protein turnover versus Sis1-Hsp70-Hsp104 mediated refolding[8]. Other chaperones have also been found at INQ including Cdc48 and the SUMO-targeting ubiquitin ligase (StUbL) Slx5. Additionally, Smt3 (SUMO) itself has been found at INQ suggesting that SUMOylation as a post-translational modification (PTM) could help govern INQ formation or degradation[16,17]. How these chaperones get recruited to INQ and how SUMOylation might impact INQ clearance remains to be understood.

Here, using Rpd3, a histone deacetylase, as an INQ marker under replication stress, we explore the role of SUMOylation at INQ. We show that perturbation of SUMOylation reduces INQ formation, while polySUMOylation plays a role in INQ clearance. In an attempt to determine whether INQ proteins are SUMOylated, we observe SUMOylation of both sHsps Btn2 and Hsp42 under DNA damage and that SUMOylation of Btn2 occurs at its C-terminus. We further map the modified lysines on Btn2 to create a non-SUMOylatable construct. We observe that Btn2 SUMOylation plays an important role in INQ clearance, is epistatic with Hsp104-mediated refolding, and independent of Apj1-assisted turnover. In addition, Btn2 SUMOylation opposes K48-ubiquitin chain formation deposited by the StUbL Slx5/8 when the Apj1 pathway is blocked, irrespective of Hsp104 status. Overall, this work reveals a DNA damage-dependent PTM of sHsps and its role in regulating the fate of these inclusions. Finally, we place Btn2 at the nexus of protein refolding and degradation at INQ, delineating the complexity of PQC at sequestration sites.

## Results

### Rpd3 is sequestered at INQ during replication stress

Previous high-throughput studies have noted relocalization of the histone deacetylases Rpd3 and Hos2 to INQ[15,17]. We confirmed this result using a C-terminal GFP fusion of Rpd3 and monitored its sequestration using fluorescence microscopy. To induce INQ formation, we used methyl methanesulfonate (MMS) which alkylates both adenine and guanine, resulting in replication fork stalling and consequently double strand breaks[20]. Following exposure to MMS, Rpd3 formed inclusions within 1 h (Fig. 1a). While Rpd3 itself is canonically present in the nucleus, we used a Hta2-mCherry nuclear marker and a Nic96-RFP nuclear periphery marker to confirm its localization (Fig. 1b, c). Indeed, we found that Rpd3 formed both nuclear and cytoplasmic inclusions upon MMS treatment. Additionally, we colocalized Rpd3-GFP with Hos2-mCherry to confirm that the Rpd3 nuclear inclusions were in fact INQ sites (Fig. 1d). While we can easily distinguish between nuclear and cytoplasmic foci, further distinction between the cytoplasmic foci (JUNQ vs IPOD) is difficult. To avoid miscalling, henceforth all cytoplasmic foci will be grouped and referred to as CytoQ inclusions.

As highlighted previously, sequestration to INQ and other inclusions is orchestrated by the chaperones Btn2 and Hsp42[12,13]. To this end, we retested Rpd3 INQ localization in *btn2Δ* and *hsp42Δ* cells. While deleting *HSP42* abrogated all inclusions, deleting *BTN2* removed INQ formation whereas CytoQ foci remained (Fig. 1e and Supplementary Fig. 1a). In line with this, cell fractionation revealed that Rpd3 was mostly soluble in unstressed cells but that induction of replication stress via MMS treatment caused a shift to the insoluble fraction (or pellet). Deleting *BTN2* reduced Rpd3 protein levels in the insoluble pellet whereas an Hsp42 deletion completely blocked the accumulation of Rpd3 in the pellet (Fig. 1f). Furthermore, the presence of some Rpd3 in the insoluble fraction in cells lacking Btn2 suggests that the pellet contained both INQ and CytoQ inclusions. While this supported the site-specific role of Btn2 targeting proteins to nuclear inclusions, it suggests that Hsp42 might also play a major role in sequestering proteins to both nuclear and cytoplasmic inclusions under DNA damage. Interestingly, Rpd3 protein levels were reproducibly lower in

*hsp42Δ* cells regardless of DNA damage stress, perhaps suggesting an unappreciated role for Hsp42 in Rpd3 expression or stability. Given the specificity of Btn2 for INQ and the lack of confounding effects on expression, we chose to focus on Btn2.

Rpd3 forms three major protein complexes that control transcriptional regulation through their histone deacetylation activity[21–23]. The Rpd3L and Rpd3S complexes both have a common catalytic core complex made up of Rpd3-Sin3-Ume1 with additional subunits that dictate complex specificity[24]. Rpd3μ on the other hand is composed of Rpd3, Snt2, and Ecm5 and is important for oxidative stress response (Supplementary Fig. 1b)[23]. Since other INQ proteins can either be sequestered within complexes (e.g., Mus81-Mms4) or can disassociate from their complexes (e.g., Hsh155), we tested whether Rpd3 complex subunits also relocalize to INQ[15,18]. We imaged C-terminal GFP fusions of ten Rpd3 complex subunits and found that none of the subunits tested relocalized to INQ upon replication stress (Supplementary Fig. 1b). This was further confirmed by fractionation of Rpd3 catalytic core subunit (Ume1), Rpd3L (Pho23), and Rpd3S (Rco1) subunits. Similar to Rpd3, a small fraction of Ume1 and Rco1 were present in the insoluble pellet in untreated cells, however, replication stress did not cause a further shift towards insolubility for the proteins tested. (Supplementary Fig. 1c). These data suggest that Rpd3 dissociates from its complexes or is sequestered to INQ as a nascent unassembled polypeptide by Btn2 and Hsp42 following MMS treatment.

### Sequestration of Rpd3 to INQ is a DNA damage response

While the major mechanism of DNA damage via MMS treatment is through the replication fork stalling, there have been reports of MMS alkylating RNA and peptides[25,26]. To understand whether INQ formation is a general response to DNA damage and not through proteotoxic artifacts introduced by MMS, we tested a panel of DNA damaging agents each having a different mechanism of action. We used hydrogen peroxide (oxidative damage)[27], camptothecin (topoisomerase-poison)[28], hydroxyurea (replication fork stalling)[29], and ethyl methanesulfonate (alkylating damage)[30] and found that all DNA damaging agents caused INQ formation for Rpd3 indicating that this is likely to be a general response to replication stress and not due to protein reactivity of MMS (Fig. 2a).

If INQ formation is a general response to replication stress, then genetic perturbations not requiring any exogenous stress should recapitulate the phenotype. To test this we used mutations that inactivate the DNA damage response kinases Mec1 and Tel1[31–33]. We found that both *mec1Δsml1Δ* and *tel1Δ* cells have elevated INQ formation after MMS treatment, hinting that a combination of MMS and a higher internal rate of DNA damage caused due to these deletions resulted in a stronger response (Fig. 2b). Importantly, a population of unstressed cells also formed INQ for Rpd3 in these mutants. Budding index analysis revealed that the cells undergoing spontaneous INQ formation were small budded cells and therefore likely to be in S-phase (Fig. 2c and Supplementary Fig. 2a). This aligns with our hypothesis that a higher burden of internal replication stress was sufficient for INQ formation as part of the DNA replication stress response.

Remarkably, *sml1Δ* cells had a similar level of INQ formation both in unstressed and MMS treated cells when compared to *mec1Δsml1Δ*, *tel1Δ*, and *mec1Δsml1Δtel1Δ* cells (Fig. 2b). Sml1 inhibits dNTP pool production and is degraded via Mec1 and Tel1-mediated phosphorylation. Consistent with our HU data, which also inhibits dNTP production, overexpression of *SML1* caused INQ formation in WT cells (Supplementary Fig. 2b). Deletions of *SML1*, rather than impairing replication, increase dNTP pools, increase replication fork speed, and favor late replication origin firing[34]. Since both increase expression and deletion of *SML1* seem to promote INQ formation, our data suggest that careful balancing of the nucleotide pool and replication dynamics are key signals for Rpd3 deposition at INQ.

To further delineate the DNA damage response arm that dictates INQ formation, we tested Rpd3-INQ formation in DNA damage checkpoint (DDC) (Rad9, Chk1) and DNA replication checkpoint (Mrc1, Tof1, Csm3) mutants[35]. Only DDC mutants had increased INQ formation suggesting that a functioning DDC opposed INQ formation during replication stress (Supplementary Fig. 2c). Overall, we concluded that sequestration of Rpd3 to INQ is a general response to DNA replication perturbations.

## Perturbing SUMOylation impacts INQ formation

Having established a system to track INQ formation and clearance through Rpd3, we sought to understand the role of PTMs in INQ regulation. Studies have revealed that ubiquitination is required for JUNQ localization but dispensable for deposition at INQ under proteolytic stress[6,7]. In contrast, ubiquitination appears to be an important sorting signal to INQ under replication stress for certain endogenous proteins[17-19]. In an effort to characterize the role of other PTMs at INQ, we focused on the small ubiquitin-like modifier, SUMO. Similar to ubiquitination, SUMOylation has well-documented roles in regulating protein stability, degradation, and subcellular localization[36-38], and previous genetic and imaging data placing yeast Smt3 at INQ, implicate SUMO in INQ regulation[16,17]. In order to probe this hypothesis, we used a ts (temperature sensitive) allele of the only SUMO E2 conjugase (UBC9) under semi-permissive temperatures where the allele is hypomorphic[39], and found that the ubc9-1 allele reduced INQ formation for Rpd3 in MMS treated cells at both 25 °C or 30 °C (Fig. 3a).

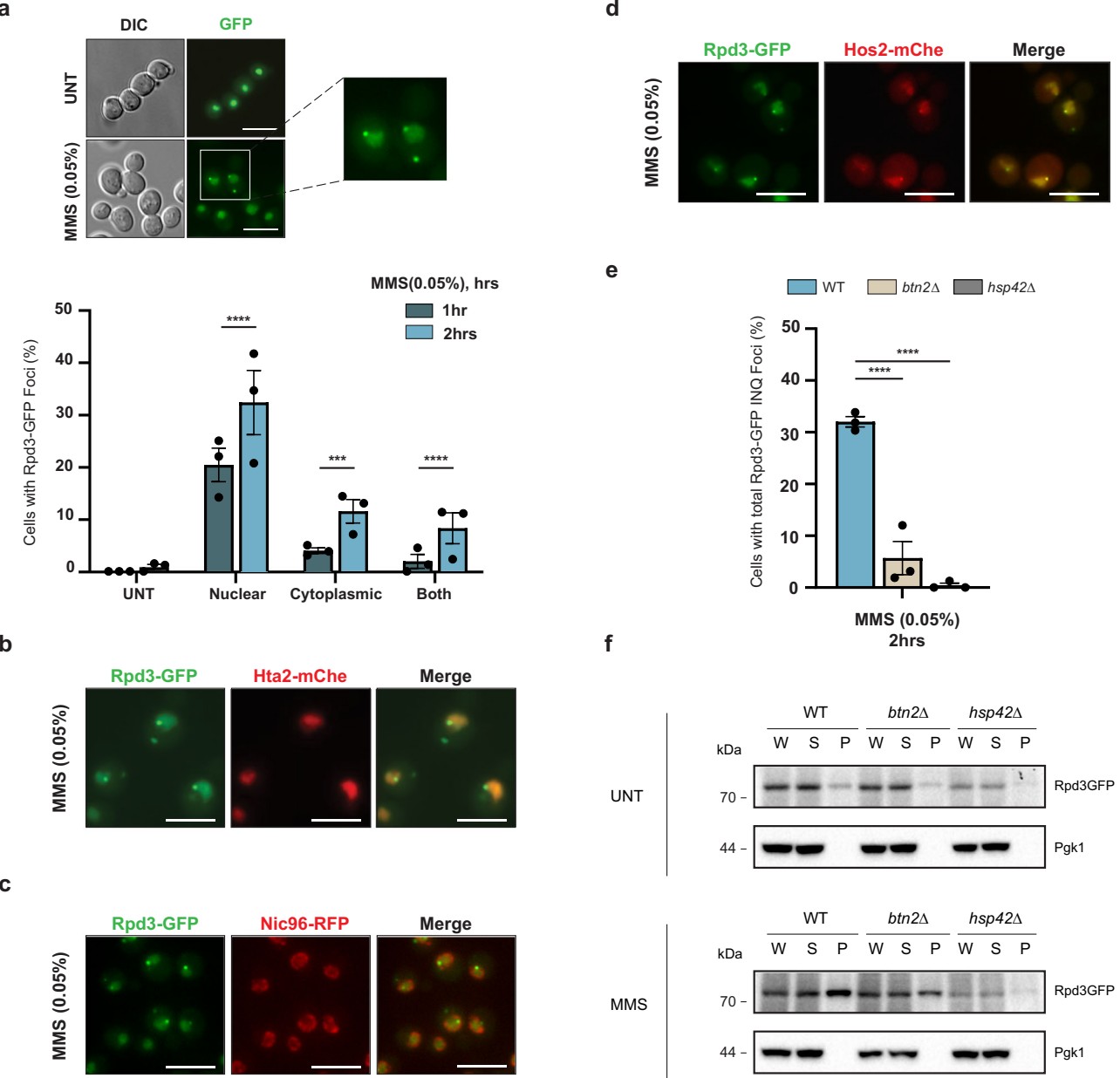

**Fig. 1 | Rpd3 is sequestered to INQ upon replication stress. a** Rpd3 forms inclusions upon acute exposure to 0.05% Methyl methanesulfonate (MMS) which induces replicative stress. Right, representative image; Left, quantification of cells with Rpd3-GFP foci. UNT = untreated. **b**–**d** Representative images of colocalization of Rpd3-GFP with Hta2-mCherry, Nic96-RFP and Hos2-mCherry respectively. **e** Deletions of compartment-specific sequestrases Hsp42 and Btn2 results in a significant reduction of Rpd3 INQ foci in MMS treated cells. **f** Cellular fractionation supports imaging data showing Hsp42 and Btn2-dependent shift of Rpd3-GFP to the insoluble pellet under stress. W whole cell extract, S supernatant, P insoluble pellet. All error bars represent means ± SEM, n = 3 biologically independent replicates, >100 cells each. ****p < 0.0001, ***p < 0.0002, **p < 0.002, Fisher's test. Scale bars: 5 μm. Source data are provided as a Source Data file.

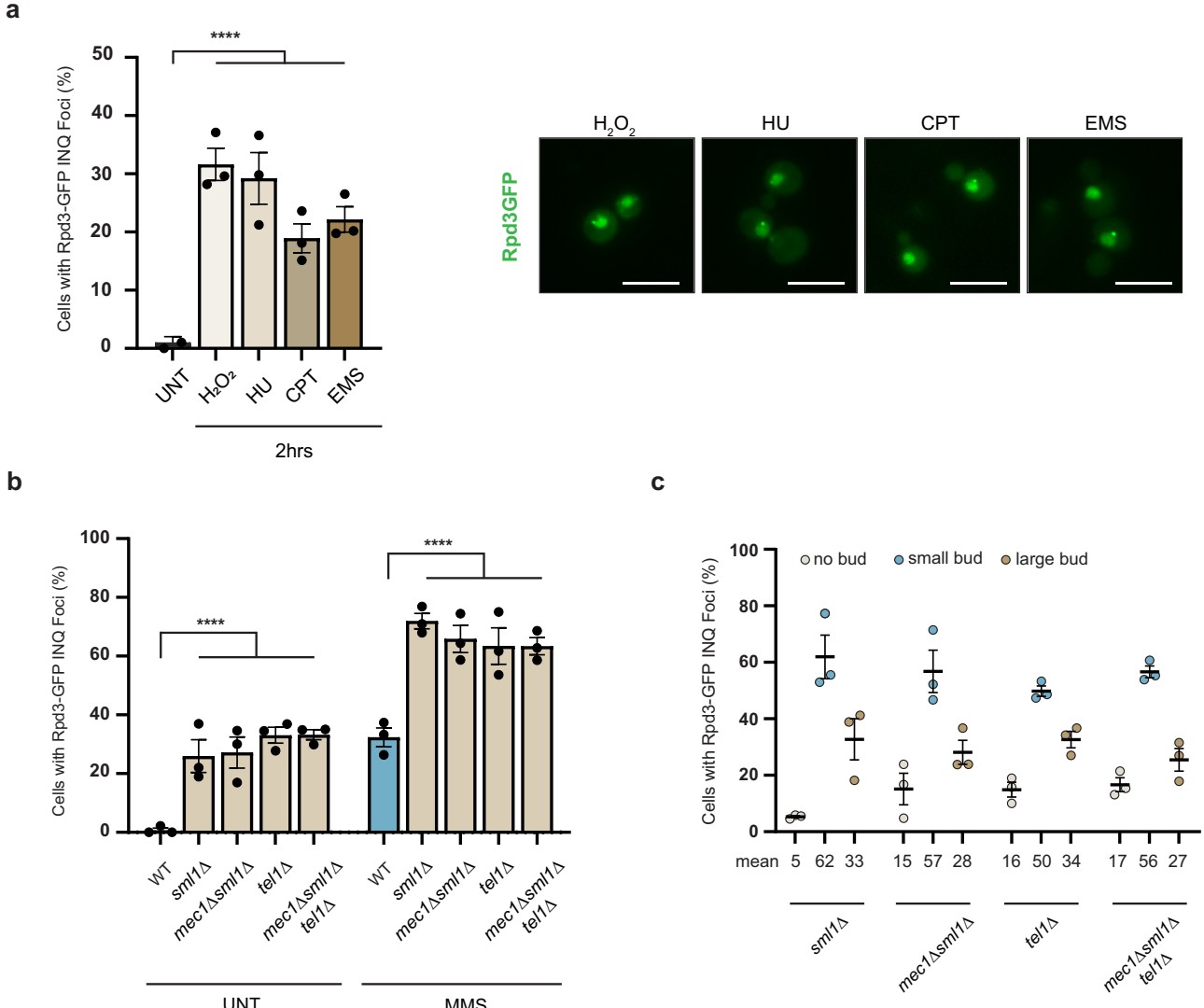

**Fig. 2 | Sequestration of Rpd3 as a general response to DNA damage. a** Rpd3-GFP foci levels following different genotoxic stressors. $H_2O_2$ (2 mM), Camptothecin (CPT; 25 uM), Hydroxyurea (HU; 200 mM) and Ethyl methanesulfonate (EMS; 0.5%). Representative images are shown below and quantified above. **b** Quantification of Rpd3-GFP INQ foci levels in *sml1Δ, mec1Δsml1Δ, tel1Δ,* and *mec1Δsml1Δtel1Δ* cells in untreated and MMS treated cells. **c** Rpd3-GFP foci counts as a function of budding index for unstressed cells harboring *sml1, mec1,* and *tel1* deletions. Mean percentage values of three replicates are shown below each dataset. All error bars represent ±SEM, *n* = 3 biologically independent replicates, >100 cells each.
****$p < 0.0001$, Fisher's test. Scale bars: 5 μm. Source data are provided as a Source Data file.

The *ubc9-1* phenotype indicated an important role for SUMOylation in INQ formation. We explored this further by creating a panel of E3 ligase deletions between Siz1, Siz2, and Mms21 owing to their redundancy between substrates. While *siz1Δ* and *siz2Δ* caused no change, cells harboring the Mms21-CH mutation[40], both on their own or combined with *siz1* or *siz2Δ* caused a reduction in Rpd3 INQ foci (Fig. 3b). Strikingly, reduction in Rpd3 INQ formation in these mutants coincided with an increase in cytoplasmic inclusion formation, suggesting that SUMOylation could be an important INQ sorting signal (Fig. 3b). In yeast, mono-SUMOylation is essential while polySUMO chains are not. We can separate their functions by working in a Smt3-3KR strain where lysines responsible for polySUMOylation are mutated to arginine[41]. Inhibiting this poly-SUMOylation process led to an increase in INQ formation for Rpd3 under MMS treatment suggesting a role for polySUMOylation in either INQ clearance or in opposing INQ formation similar to the checkpoint and kinase mutants in Fig. 2 (Fig. 3c). DNA damage is known to induce a wave of SUMOylation that aids in DNA repair and stress response[42,43]. Indeed we saw that DNA damage also induced SUMOylation of the pellet and this coincided with increased ubiquitination, including protein degradation linked-K48 ubiquitin chains, in MMS-treated cells (Fig. 3d). Since poly-SUMOylation and subsequent ubiquitination governs degradation of many proteins[44], we retested fractionation in these cells and found a slight reduction in K48-Ubiquitin chains and more SUMOylated proteins in the insoluble fraction supporting a role for polySUMOylation in degradation of proteins at INQ (Fig. 3d).

In conclusion, mutations perturbing SUMOylation reduced INQ formation for Rpd3 under replication stress. While combining E3 ligase mutations and deletions, especially containing the Mms21-CH allele, caused a reduction in INQ formation, they concomitantly caused an increase in CytoQ foci indicating a role for SUMOylation as a sorting signal alongside ubiquitination between sequestration compartments. Furthermore, while SUMOylation might be necessary for INQ formation, polySUMOylation might have a role in INQ clearance hinting towards a complex stepwise role for SUMOylation at INQ.

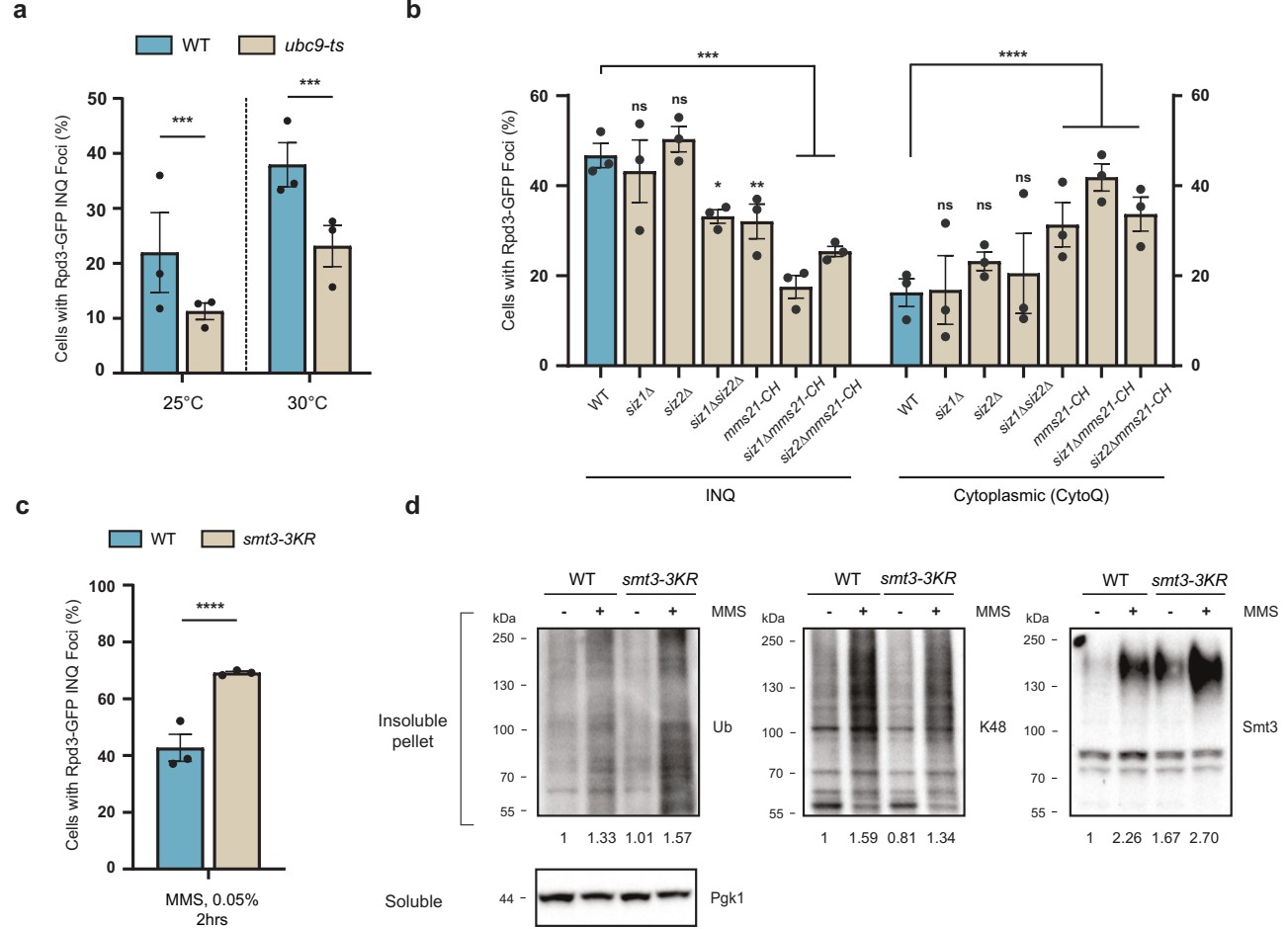

**Fig. 3 | DNA damage induced SUMOylation regulates INQ levels. a** Rpd3 foci levels in a temperature sensitive allele of the SUMO E2 conjugase Ubc9 at the indicated temperatures in MMS. **b** Rpd3-GFP foci in the nucleus and cytoplasm of strains with the indicated SUMO E3 ligase mutations in MMS treated cells. **c** Rpd3-GFP foci levels when blocking polySUMOylation using the Smt3-3KR allele. **d** Fractionation in WT and Smt3-3KR cells with and without MMS reveals a wave of SUMOylation and ubiquitination upon DNA damage. Total ubiquitin (Ub), K48-Ubiquitin, and SUMO levels quantified with respect to WT untreated cells are indicated below the blot. All error bars represent ±SEM, $n = 3$ biologically independent replicates, >100 cells each. ****$p < 0.0001$, ***$p < 0.0002$, **$p < 0.002$, *$p < 0.03$, ns, $p > 0.1$, Fisher's test. Source data are provided as a Source Data file.

## SUMOylation of sequestrase chaperones under DNA damage

Our results indicated that SUMOylation controls INQ deposition, suggesting that DNA damage might also induce SUMOylation of proteins that reside at INQ. Previous direct tests of INQ substrates Hsh155 or Mrc1 under MMS treatment did not detect any modification[16,45]. We sought to expand the set of INQ clients in the hopes of finding SUMO modifications in a more comprehensive manner. We chose five substrates, splicing factors Hsh155[16] and Cdc40, Rpd3 and Hos2, and Cmr1, and five chaperones, sHsps Btn2 and Hsp42, Apj1 (Hsp40), Hsp104 and Cdc48 (Fig. 4a). To ensure uniformity between studies detailing INQ formation for these proteins, we chose GFP fusions for this mini-screen. Additionally, we used DNA damage-dependent SUMOylation of Rfa1 as a positive control for both DNA damage and a successful pulldown of purified SUMO proteins using the established Smt3-Hisx7 nickel bead denaturing pulldown[16,46].

While MMS treatment caused clear mono-, di- and tri-SUMOylation of Rfa1 in cells expressing the Smt3-Hisx7 fusion (Fig. 4b), we found that the five substrates tested were not SUMOy-lated (Supplementary Fig. 3a). On the contrary, we noted clear SUMOylation of both sHsp sequestrases Btn2 and Hsp42 but for none of the other chaperones tested under DNA damage (Fig. 4b and Supplementary Fig. 3b). Btn2 functions as an INQ-specific sequestrase whereas Hsp42 also dictates CytoQ formation[7]. To try to understand

the impact of sequestrase SUMOylation on INQ specifically, we focused our follow-up studies on mapping and testing the function of Btn2 SUMO sites rather than Hsp42 which has more pleiotropic effects on the cell. To first confirm the result we again used the *ubc9-1* allele and found that Btn2 could not be SUMOylated at either 25 °C or 30 °C (Fig. 4c), confirming the allele is hypomorphic (Fig. 3a) and that Btn2 is SUMO modified. To date, genetic data indicates that Btn2 is important to drive INQ formation and we hypothesize that SUMOylation could either support or oppose this role. To test this, we needed to create alleles of Btn2 with reduced SUMOylation following DNA damage.

Similar to other sHsps, Btn2 contains a characteristic alpha-crystallin domain (ACD) flanked by N-terminal and C-terminal domains (NTD and CTD, respectively). The NTD contains the nuclear localization signal and is also responsible for Sis1 (and therefore Hsp104) recruitment while the CTD is highly disordered and appears to be involved in INQ formation through a mechanism that is unclear[13]. In order to study these protein regions in the context of the SUMOylated domain, we created three constructs: *wt* (*BTN2* full length), *ctdΔ* (C-terminal domain deleted), and *ctd+cldΔ* (only the NTD) (Fig. 4d). Using pulldown of Smt3-Hisx7, only the full-length *wt* construct was SUMOylated under DNA damage indicating that SUMOylation likely occurred within the CTD (Fig. 4e and Supplementary Fig. 3c). We also fused a GFP at the N-terminus since an N-terminal GFP fusion for Btn2

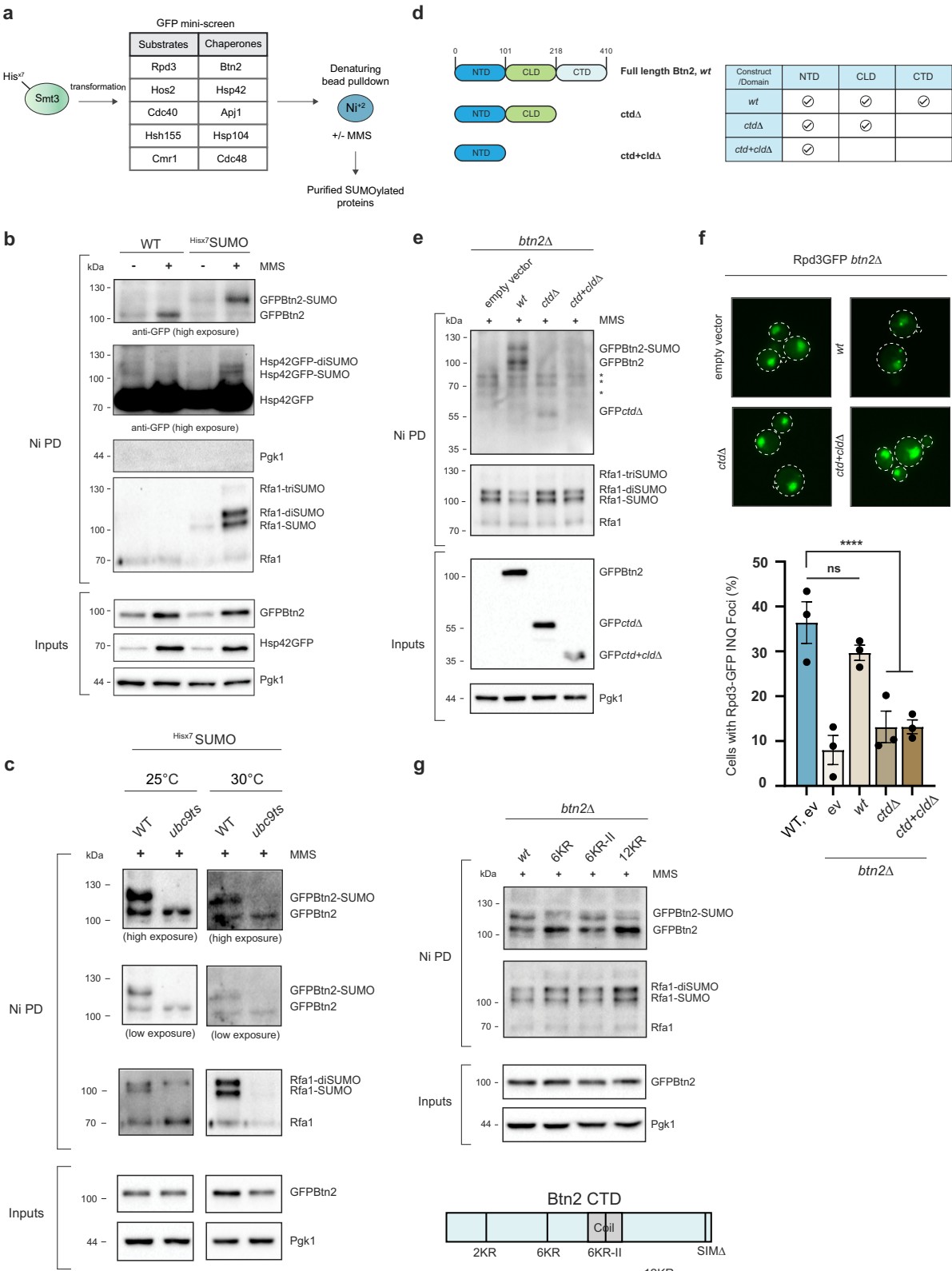

**Fig. 4 | DNA damage induces SUMOylation of sHsps Btn2 and Hsp42.**
**a** Schematic of the SUMOylation mini-screen utilizing GFP-constructs of substrates and chaperones found at INQ. **b** MMS treatment induces SUMOylation of Btn2 and Hsp42. Shown is a nickel bead denaturing pulldown (NiPD) of His-SUMO conjugates in Hsp42-GFP and GFP-Btn2 backgrounds and western blots for GFP, PGK1 or RFA1. **c** Mutating the E2 SUMO conjugase using the *ubc9-1* temperature-sensitive allele at semi-permissive temperatures removes Btn2 SUMOylation. NiPD was performed at both 25 °C and 30 °C. **d**, **e** Schematic and NiPDs of GFP-Btn2 constructs used to

determine the SUMOylated region reveal CTD as the SUMOylated domain. * represents non-specific banding pattern as noted in all wells. **f** Rpd3 INQ foci rescue experiments in *btn2Δ* using GFP-Btn2 domain constructs. Representative images are shown and quantified below. **g** Testing Btn2 CTD lysine-to-arginine constructs reveal Btn2-6KR SUMOylation is significantly reduced upon DNA damage. NiPDs presented in (**b**, **c**, **e** and **g**) were conducted in duplicates. All error bars represent ±SEM, *n* = 3 biologically independent replicates, >100 cells each. ****ns, *p* > 0.1, Fisher's test. Scale bars: 5 μm. Source data are provided as a Source Data file.

appeared more functional compared to the C-terminal fusion based on the literature[47]. Indeed, GFP-Btn2 was nuclear in the *wt* construct and formed spontaneous nuclear inclusions that appeared to be INQ (Supplementary Fig. 3d). This full-length construct also rescued Rpd3 sequestration to INQ in a *btn2Δ* background indicating that the construct was functional (Fig.4f). Consistent with literature[13], the *ctdΔ* failed to rescue INQ formation for Rpd3 suggesting a functional role in sequestration. However, this could be due to mislocalization of Btn2 since the GFP-Btn2 *ctdΔ* also formed cytoQ inclusions (Supplementary Fig. 3d). Lastly, similar to *ctdΔ*, the *ctd+cldΔ* construct also failed to rescue INQ formation which was consistent with the requirement of an ACD to bind substrates. Since the ACD is deleted in the *ctd+cldΔ* construct, Btn2 failed to form any nuclear inclusions (Fig. 4f and Supplementary Fig. 3d). These data support key roles for the chaperone and CTD of Btn2 and implicate the CTD as a candidate for MMS-induced SUMOylation.

To reaffirm CTD SUMOylation, we mutated all CTD lysines to arginine to abolish SUMOylation (CTD-KallR). Consistent with the hypothesis, the *wt* construct was SUMOylated under DNA damage while the full length GFP-Btn2-CTD-KallR was not (Supplementary Fig. 3e). Since the CTD-KallR was highly mutated (26 lysines mutated to arginine), we sought to create a minimally mutated Btn2 to test the function of SUMOylation. To this extent, we used the SUMO-prediction software JASSA to create four different constructs of the CTD where nearby lysines were mutated to arginine to avoid jumping of the modification[48,49]. This approach created the following constructs: 2KR (predicted site−K234), 6KR (predicted site−K249), 6KR-II (predicted site−K290), and a 12KR construct (no high confidence predictions in this region). We repeated the nickel bead pulldowns under MMS to reveal that out of the four constructs created, only Btn2-6KR showed significant loss of Smt3 signal (Fig. 4g and Supplementary Fig. 3f). To avoid potential pitfalls with a plasmid system for genetic studies, we introduced these mutations to the genome to create either Btn2-WT or Btn2-6KR with a URA3 selection marker upstream of Btn2. Loss of SUMOylation was reassessed and confirmed after integration (Supplementary Fig. 3g). It is important to note that we observe a weaker DNA-damage insensitive SUMOylation in Btn2-6KR (quantified in Supplementary Fig. 3g). We know this is likely SUMO because it was abolished in *ubc9-1* alleles, and suggests that Btn2 may be SUMOylated constitutively at some level on other sites (Fig. 4c). Indeed, we also note a low level of SUMOylation for Btn2 in untreated cells (Fig. 4b). Importantly, Btn-6KR reduces DNA damage induced SUMO by almost 3-fold (Supplementary Fig. 3g) without affecting Btn2 protein levels under MMS, making it a useful tool to probe the impact of this PTM. In summary, we reveal SUMOylation of the sHsps Btn2 and Hsp42 under DNA damage. Using a domain-specific approach, we found that Btn2 was SUMOylated at its CTD in the K249 region, a domain that appears to be important for INQ formation.

## Non-SUMOylatable Btn2 is a functional sequestrase

Having engineered Btn2-6KR as an allele with minimal residual SUMOylation we sought to characterize its effects on INQ biology. Since SUMOylation occurred under DNA damage, we first tested growth under varying MMS concentrations and temperatures but observed no major difference between Btn2-WT and Btn2-6KR (Supplementary Fig. 4a). We also tested recovery from a two-hour MMS treatment and found that the strains recovered at similar rates (Supplementary Fig. 4b). Next, we focused on Btn2 localization. SUMOylation has been shown to govern localization for various proteins such that mutation of the modified lysine causes mislocalization to a different subcellular compartment[36]. Both Btn2-WT and Btn2-6KR localized to the nucleus irrespective of SUMOylation status (Fig. 5a).

SUMOylation is also directly linked to protein degradation as it can act as a signal for ATPases such as Cdc48 that target SUMOylated proteins for degradation[50]. Moreover, SUMOylation can serve as a platform for StUbLs that can ubiquitinate SUMOylated proteins for degradation[37,51]. To test whether SUMOylation status of Btn2 affects its stability, we performed a cycloheximide (CHX) chase experiment. Additionally, we combined the CHX chase with fractionation to test whether SUMOylation affects protein stability in the pellet vs the soluble fraction. SUMO-defective Btn2-6KR appeared to have a similar stability to Btn2-WT (Supplementary Fig. 4c). This was recapitulated in the soluble and pellet fraction with Btn2 degrading at the same rate. Moreover, isolation of SUMO-modified Btn2 in a *cim3-1* proteasome-defective mutant at non-permissive temperatures of 30 °C and 37 °C showed no major change in SUMO-modified Btn2 suggesting that the SUMOylation status of Btn2 is not associated with its degradation (Supplementary Fig. 4d).

While fractionating cells in the CHX chase experiments, we noticed that Btn2-6KR was more abundant in the insoluble fraction than Btn2-WT (Fig. 5b). Loss of SUMOylation appeared to make Btn2 more insoluble, consistent with the role of SUMO as a solubility tag[52–54]. This was further assessed by performing a filter trap assay on the fractionated samples where more GFP-Btn2-6KR was retained compared to GFP-Btn2-WT for the same amount of protein loaded confirming that excess Btn2-6KR is retained in the insoluble state (Fig. 5c). To attribute the increase in insolubility to loss of SUMOylation and not the mutations, we used alphafold2 to predict structures for Btn2-WT and Btn2-6KR. These were fed to two softwares CamSol and Aggrescan 3D which predict protein solubility and aggregation potential per residue. When comparing Btn2-WT and Btn2-6KR in the region specific to the mutations (represented in the bin) the mutations to arginine were actually predicted to make the binned region more soluble (Supplementary Fig. 4e). Furthermore, using FoldX, we calculated the predicted energy difference ($\Delta\Delta G$) between Btn2-WT and Btn2-6KR to be −0.2 kJ/mol which is far lower than $\Delta\Delta G$ values for destabilizing mutations of >1 kJ/mol[55]. These analyses support our contention that the mutations in Btn2-6KR did not destabilize or affect protein solubility (Supplementary Fig. 4e). Rather the change in solubility could be attributed to loss of SUMOylation. Overall, non-SUMOylatable Btn2-6KR appears to be as stable as Btn2-WT to localize normally and to permit cellular fitness after DNA damage and enable INQ formation. However, Btn2-6KR does appear more insoluble following DNA damage suggesting potential differences in the state of INQ or its lifetime.

## Btn2-SUMO restrains INQ accumulation

Next, we sought to understand the role of Btn2 SUMOylation at INQ. Since Btn2-6KR was more insoluble, we wondered whether INQ as a compartment would also be more insoluble. To test this, we chose four substrates (Rpd3, Hos2, Hsh155, Cmr1) and four PQC components (Cdc48, Slx8, Apj1, Hsp104). We included Slx8 here as a proxy for the Slx5/8 complex that acts as a StUbL to degrade SUMOylated proteins[56,57]. In order to confirm INQ localization, all proteins were colocalized with Hos2-mCherry under MMS treatment. This analysis showed that loss of Btn2 SUMOylation in Btn2-6KR resulted in elevated sequestration of INQ substrates Rpd3, Hos2, Hsh155, and Cmr1, and PQC factors Cdc48 and Slx8 (Fig. 5d). Fractionation for Rpd3 further supported this data, as Rpd3 was more insoluble in Btn2-6KR compared to Btn2-WT after MMS treatment (Fig. 5e). These data suggest that SUMOylation of Btn2 is not the signal required for sequestration or INQ formation since it appears to be unaffected in Btn2-6KR. Interestingly, we found a small yet significant decrease for localization of proteins Apj1 and Hsp104, both of which have well established roles in INQ clearance (Fig. 5d). This indicated that Btn2 SUMOylation and therefore INQ solubility might play a role in INQ clearance and recovery by affecting one or both of these pathways.

Elevated sequestration of INQ proteins in Btn2-6KR cells suggested a defect in INQ clearance or recovery. To support this notion, we utilized the heat stress sensitive growth phenotype of cells with

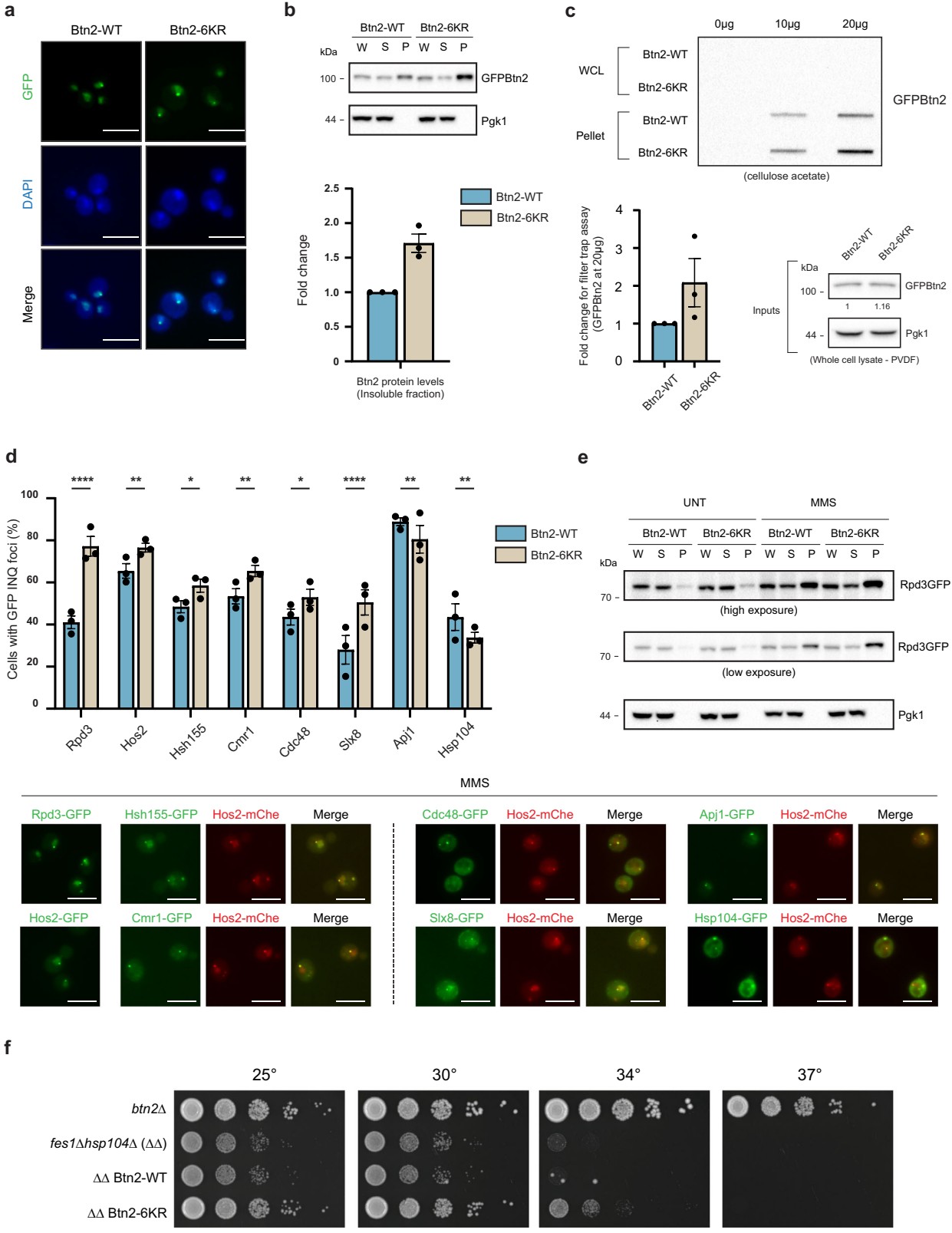

*fes1Δhsp104Δ* that is caused due to a high burden of misfolded proteins[13]. It has been shown that deletion of *BTN2*, blocking sequestration in the *fes1Δhsp104Δ* background exacerbates fitness defects, supporting the important role for INQ in survival. Remarkably, and consistent with an important role for INQ sequestration, the elevated levels of INQ in Btn2-6KR actually correlated with partial rescue of temperature sensitivity in *fes1Δhsp104Δ* (Fig. 5f). Thus, Btn2

SUMOylation may be important for INQ dissolution, but a hyper-sequesteration phenotype can actually be beneficial in chaperone-deficient genetic backgrounds.

## Btn2 SUMOylation regulates INQ clearance
Two established and independent pathways that govern INQ clearance are the Hsp104-mediated protein refolding pathway along with the

**Fig. 5 | Btn2 SUMOylation restricts INQ accumulation. a** Localization of Btn2 is unaffected by the 6KR mutation. **b** Fractionation of Btn2-WT and Btn2-6KR cells reveals increased insolubility for Btn2-6KR in the pellet fraction. Shown is a graph measuring relative fold change for pellet fraction Btn2 compared to WT from three independent experiments. **c** Filter trap assay performed using a cellulose acetate membrane retained more Btn2 in Btn2-6KR cells indicating increased insolubility due to loss of SUMOylation. Relative fold change was calculated between equal amounts (20 μg) of protein loaded from three independent experiments (lower left panel). Equal amounts of whole cell lysates were run on an SDS-PAGE and quantified to control for protein levels (lower right panel). Relative fold change was calculated from five independent experiments. **d** Loss of Btn2 SUMOylation causes elevated sequestration for multiple INQ proteins. Representative images are shown below. **e** Fractionation of Rpd3 in both Btn2-WT/Btn2-6KR strains in untreated and MMS treated cells. **f** Btn2-6KR growth effects in a *fes1Δhsp104Δ* background as seen from a spot dilution assay. Equal ODs of the indicated strains were serially diluted and spotted on YPD at 25 °C, 30 °C, 34 °C and 37 °C. All error bars represent ±SEM, *n* = 3 biologically independent replicates, >100 cells each. ****$p$ < 0.0001, **$p$ < 0.002, *$p$ < 0.03, Fisher's test. Scale bars: 5 μm. Source data are provided as a Source Data file.

Apj1-mediated degradation pathway[58]. We sought to determine whether Btn2-6KR induced insolubility is epistatic with either of the two or is a third pathway mediating clearance. We treated cells harboring Btn2-WT or Btn2-6KR combined with *apj1Δ*, *hsp104Δ*, or *apj1Δhsp104Δ* with MMS for two hours followed by a washout into MMS-free media and studied the rate of clearance for Rpd3 INQ foci. In WT cells, we noted that despite Btn2-6KR having a higher percentage of Rpd3 INQ foci than Btn2-WT, both strains successfully cleared INQ such that after 2 h only 20% of cells harbored INQ (Fig. 6a). In contrast, while *apj1Δ* Btn2-WT cells could clear INQ, *apj1Δ* Btn2-6KR cells were completely defective in clearance during the time course of this experiment and instead continued to buildup Rpd3 foci (Fig. 6a). Therefore, preventing Apj1-mediated degradation and concomitantly blocking Btn2 SUMOylation resulted into a complete inability to clear INQ in the time course of this experiment (Fig. 6a).

Since Apj1 and Hsp104 execute independent pathways, we hypothesized that Btn2-6KR and Hsp104 should be epistatic. Indeed, while slower than WT, both *hsp104Δ* Btn2-WT and *hsp104Δ* Btn2-6KR cells had lower INQ foci levels at the end of a 2-h washout (Fig. 6a). Interestingly, deleting both Apj1 and Hsp104 in Btn2-WT cells blocked INQ clearance and further removing Btn2-SUMOylation did not exaggerate this phenotype (Fig. 6a). We noted the same results when we compared Rpd3 fluorescence intensities at INQ foci across different genotypes (Fig. 6b). Furthermore, studying cytoplasmic inclusion clearance revealed that while *apj1Δ* Btn2-6KR cells cleared these inclusions slower compared to *apj1Δ* Btn2-WT cells, these cytoplasmic sites were eventually cleared indicating an INQ-specific defect (Supplementary Fig. 5a). In addition, we also noted an increase in cells containing ≥2 cytoplasmic foci in Btn2-6KR cells regardless of chaperone status (Supplementary Fig. 5b) similar to Hsp42 mutants as noted in another study[12]. However, the physiological relevance of this phenotype remains to be investigated.

Since *apj1Δhsp104Δ* cells had defective INQ clearance that could not be exacerbated with a non-SUMOylatable Btn2, we wondered whether there were differences in the SUMO and ubiquitin status of these inclusions. To test this, we performed cellular fractionation and probed for total SUMO, ubiquitin, and the degradation-specific K48 ubiquitin chains. As expected, *apj1Δ* cells had an accumulation of total and K48-linked ubiquitinated and SUMOylated proteins, and further deleting Hsp104 did not exacerbate this phenotype (Fig. 6c). Importantly, the ability to SUMOylate Btn2 impacted the ubiquitin status of the fractionated inclusions. While Btn2-6KR itself did not cause any change, *apj1Δ* Btn2-6KR cells had higher total and K48-linked ubiquitin compared to *apj1Δ* Btn2-WT cells (Fig. 6c). We also noted a similar increase for K48-Ubiquitin in *apj1Δhsp104Δ* Btn2-6KR cells. This strongly suggested a role for Btn2 SUMOylation in negatively regulating the build-up of K48-Ubiquitin chains in *apj1Δhsp104Δ* inclusions.

### Slx5/8 activity on the MMS-induced insoluble proteome

Finally, the results suggest the action of a K48-Ubiquitin ligase at MMS-induced inclusions that is involved with Apj1. Slx5/8 is a good candidate for this activity since it is a StUbL that localizes to INQ but its effect on INQ SUMOylation and ubiquitination is unknown (Fig. 5d). To investigate this, we first tested Btn2 SUMOylation in *slx5Δ* cells. Deleting *SLX5* increased Btn2 SUMOylation, introducing visible poly-SUMO chains that were not observed in WT conditions, suggesting a constant action of either StUbLs or SUMO proteases at INQ maintaining polySUMOylation Btn2 (Fig. 6d). In agreement with our previous results in Fig. 5d, loss of Btn2 SUMOylation resulted in elevated sequestration in the backgrounds tested, indicating that Btn2 SUMOylation is not the signal for recruitment of Slx5/8 to INQ. (Fig. 6e and Supplementary Fig. 5c). Interestingly, deletion of *APJ1* significantly increased Slx8 foci suggesting that the increase in pellet ubiquitination in these cells might be due to the activity of the Slx5/8 complex.

To this end, we tested ubiquitination and SUMOylation status of the pellet proteome in *apj1Δ*, *hsp104Δ*, and *apj1Δhsp104Δ* cells by combining it with an *slx5Δ*. Consistently, we found that accumulation of K48-Ubiquitin chains in *apj1Δ* and *apj1Δhsp104Δ* cells was dependent on Slx5 (Fig. 6f). Contrary to previous results stating Apj1 and Slx5 are parallel pathways for degradation of SUMOylated substrates, these results place Slx5/8 in the Apj1-mediated degradation pathway at INQ as the E3 ligase responsible for ubiquitinating proteins. To further support this hypothesis, we performed MMS washouts in *slx5Δ* Btn2-WT and *slx5Δ* Btn2-6KR cells. Coupling *slx5Δ* with Btn2-6KR resulted in a totally defective INQ clearance while not affecting CytoQ clearance, similar to *apj1Δ* Btn2-6KR cells (Fig. 6g and Supplementary Fig. 5d).

In conclusion, we place Btn2 SUMOylation at the core of the triage decision at INQ. We show that while Btn2 SUMOylation is epistatic with Hsp104-mediated refolding, it appears to be parallel to Apj1-mediated degradation. Additionally, we utilize fractionation to show that Btn2 SUMOylation negatively regulates K48-Ubiquitin chain building in the MMS-induced insoluble proteome when Apj1 is absent. Furthermore, we show that contrary to previous studies showcasing Apj1 and Slx5 to be parallel pathways for degradation of SUMOylated proteins, we showcase the role of the Slx5/8 complex as the primary E3 ligase in *apj1Δ* cells. Surprisingly, localization of Slx8-GFP in *apj1Δ* is unaffected of Btn2 SUMOylation status. This is intriguing as we still note increased ubiquitination in *apj1Δ btn2-6KR* cells. One explanation could be that Btn2 SUMOylation acts as a primary substrate for Slx5/8, thus shielding other proteins at INQ from ubiquitination. However, upon loss of Btn2 SUMOylation, Slx5/8 can ubiquitinate substrates more freely. This could also explain why we do not notice increased ubiquitination upon MMS treatment in WT cells since Slx5/8-ubiquitinated proteins are triaged by Apj1 for degradation. While this could explain one such phenotype, additional work will be required to delineate the intricate crosstalk between the PQC pathways at INQ.

## Discussion

Rpd3 was previously identified as a candidate INQ resident protein in a large-scale screen[17] but this study presents the first direct tests of Rpd3 as an INQ resident protein. We show that Rpd3, but not its normal interaction partners, becomes insoluble and INQ-associated upon replication stress. This behavior is regulated by chaperone networks previously linked to INQ. Interestingly, Rpd3 shares a lysine deacetylase activity with another INQ protein, Hos2, both of which are known histone deacetylases[59]. Why these deacetylases relocate to INQ during stress remains unknown. Rpd3 and Hos2 have many substrates and

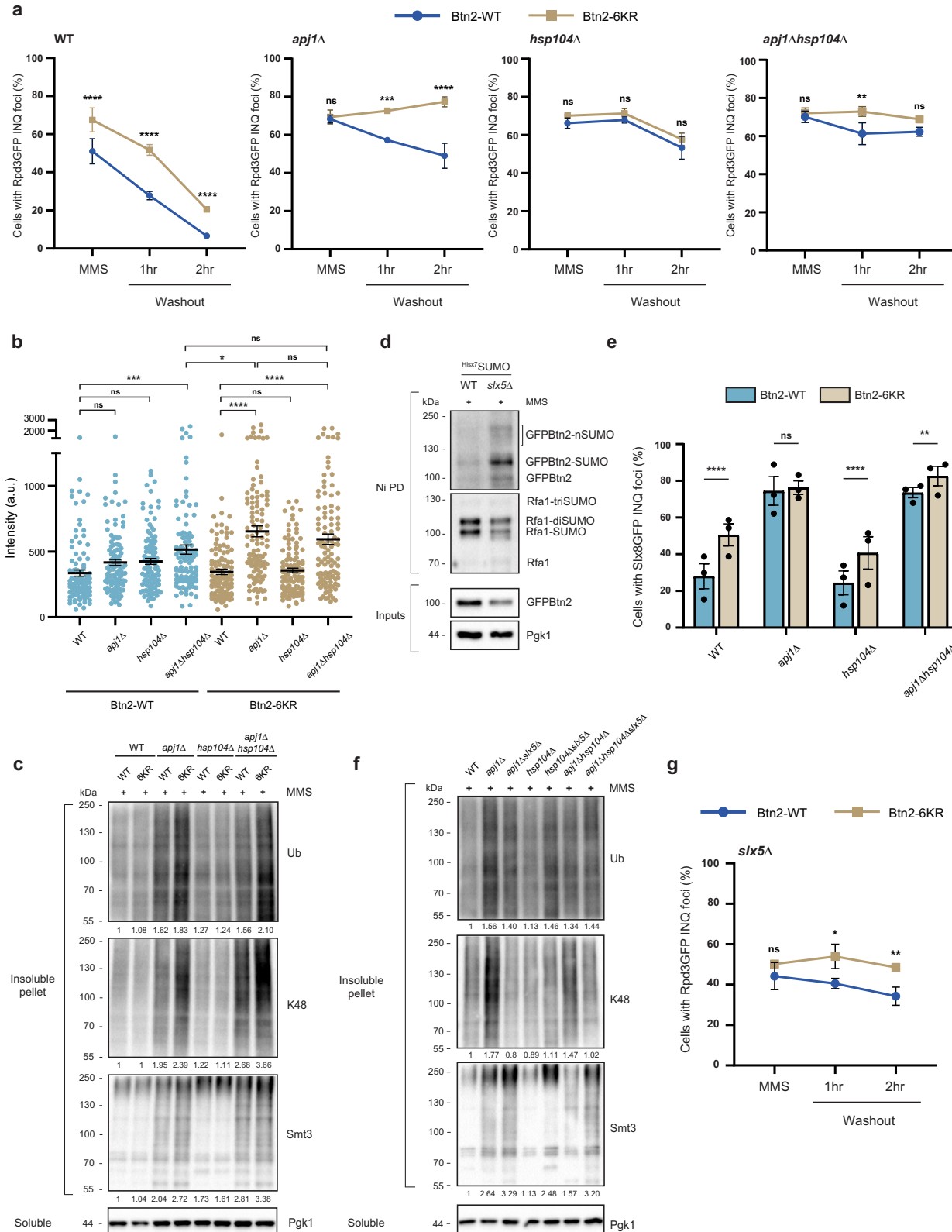

large scale effects on gene expression and on genome stability[60,61]. It is possible that the dramatic gene expression reprogramming that takes place upon MMS treatment requires immediate post-transcriptional control of HDAC activity that is achieved through sequestration. This would be similar to a model we previously presented for the INQ localization of a core splicing factor, Hsh155, whose sequestration could reduce splicing flux of ribosomal protein genes during the stress response[15]. It is also possible that roles for Rpd3 in DNA replication must be regulated during the MMS response. Rpd3 has been implicated in delaying the firing of late replicating origins[62], and we note spontaneous recruitment of Rpd3 to INQ in *sml1Δ* cells that have faster replication velocity, and early firing of late replicating origins[34]. Thus, Rpd3 sequestration at INQ could help rapidly change the chromatin state at late origins, promoting recovery from MMS. Indeed, this kind

**Fig. 6 | Btn2 SUMOylation opposes K48-Ubiquitin chain buildup when INQ-degradation is blocked. a** WT, *apj1Δ*, *hsp104Δ*, and *apj1Δhsp104Δ* cells were treated with MMS for two hours followed by a washout into MMS-free media. Shown are Rpd3-GFP foci percentages in these backgrounds after 2 h of treatment and after one and two hours of washout from three independent biological replicates per time point. **b** Intensities of INQ foci were measured for strains in (**a**) and both sets of data reveal epistasis between Btn2 SUMOylation and Hsp104-mediated refolding while indicating that these two pathways may operate in parallel to Apj1-mediated degradation. **c** Fractionation in WT, *apj1Δ*, *hsp104Δ*, and *apj1Δhsp104Δ* cells treated with MMS and western blot for levels of total Ubiquitin (Ub), K48-linked Ubiquitin chains, and SUMO/Smt3 chains. **d** Deleting Slx5 causes an increase in Btn2 mono-SUMOylation and polySUMOylation. NiPD was performed in WT and *slx5Δ* cells treated with MMS. **e** Slx8-GFP INQ foci levels in WT, *apj1Δ*, *hsp104Δ*, and *apj1Δhsp104Δ* cells with Btn2-WT/Btn2-6KR following MMS treatment. **f** Fractionation in WT, *apj1Δ*, *hsp104Δ*, and *apj1Δhsp104Δ* cells with or without *slx5Δ*. All cells were treated with MMS and probed for levels of total Ub, K48-Ub chains, and Smt3 chains. **g** Rpd3-GFP foci percentages in *slx5Δ* cells harboring Btn2-WT/Btn2-6KR following the same experiment design and washout in (**a**). Total ubiquitin, K48-Ubiquitin, and SUMO levels quantified with respect to Btn2-WT MMS cells are indicated below the blot for (**c**, **f**). All error bars represent ±SEM, $n = 3$ biologically independent replicates, >100 cells each. ****$p < 0.0001$, ***$p < 0.0002$, **$p < 0.002$, *$p < 0.03$, ns, $p > 0.1$, One-way ANOVA for (**b**), Fisher's test for (**a, e, g**). Source data are provided as a Source Data file.

of adaptive sequestration to INQ has been proposed for Mus81 and other DNA endonucleases, where the authors suggested INQ as a regulatory hub for controlling the timing of nuclease activity late in the cell cycle[18]. Additional work is required to understand the functional significance and substrate selection criteria for novel INQ proteins like Rpd3.

The potential role of SUMO modification at INQ was first identified in 2015 with the discovery of Smt3 localization to INQ during stress[17]. SUMO modifications are known to alter the solubility of target proteins and are increasingly appreciated as regulators of membraneless phase-separated compartments (Reviewed in ref. [63]). Our study identifies targets for SUMO modification at the INQ for the first time. Surprisingly, we found that it is small heat shock proteins (sHsps), not INQ client proteins, that were SUMO modified in our screen. Our data suggest that Btn2 SUMOylation is dispensable for INQ formation and instead impacts solubility and recovery processes. sHsps bound to non-native client proteins and are thought to exist in a substructured system—a stable inner core with few sHsps bound to substrates and a dynamically exchanging sHsp shell that enables access of refolding or degradation machinery to substrates[64]. Our data suggests that SUMOylation might modify the solubility or accessibility of substrate proteins sequestered by Btn2 to other PQC machinery that enables dissolution of INQ upon stress removal. Cells unable to SUMOylate Btn2 promote a less-soluble INQ state that is incapable of dissolving Rpd3 at INQ in the absence of the Apj1 co-chaperone, which has been linked to INQ protein degradation[58]. Our data on INQ clearance and abundance following washout of MMS supports the view that Apj1 and Btn2-SUMO work in parallel to promote turnover of proteins from INQ. Additionally, loss of Btn2 SUMOylation might impair the ubiquitin-proteasome system (UPS) mediated degradation of specific substrates, since published work indicates that impaired UPS mutants also rescue the *fes1Δhsp104Δ* growth defect similar to Btn2-6KR[65]. Indeed, we observed that cells with non-SUMOylatable Btn2 accumulate K48-linked ubiquitin-modified proteins in the insoluble fraction when Apj1 was absent. K48-Ubiquitin chains on INQ proteins only accumulate in cells with functional Slx5, a SUMO targeted E3 ubiquitin ligase with known roles in genome maintenance[41,66]. When Btn2 cannot be SUMOylated the Slx5/8 complex seems to localize in excess to INQ, and further loss of Apj1 reveals K48-Ubiquitin chain accumulation and delayed recovery. These data support a model in which either Slx5/8 is hyperactive without Btn2-SUMO, or that Slx5/8 substrate proteins at INQ cannot be extracted efficiently, slowing K48-modified substrate removal and INQ recovery (Fig. 7).

In the future, it will be important to determine the function and contribution of Hsp42 SUMOylation that we discovered to this process. Only a minor fraction of Hsp42 appears to be modified, which could suggest a functional role in aiding sHsp-oligomer formation wherein only a small fraction of Hsp42 SUMO could act as a seed for oligomerization or in regulating specific protein-protein interactions. Alternatively, there may be other stress conditions in which Hsp42 is highly modified. Since Hsp42 plays important roles at cytoplasmic

quality control compartments, it is possible that SUMOylation will have more complex roles in regulating triage to these various sites under different conditions. High-throughput studies have reported Hsp42 SUMOylation on lysines 186 and 346, and it will be important for future work to investigate whether these sites are the same as that modified in our study[67].

It is important to note that our study investigates the role of SUMOylation at INQ under the context of DNA damage. Other studies have shown that stresses such as heat shock and proteasomal impairment also result in sequestration of proteins to INQ. Under these conditions, Btn2 and Hsp42 have been shown to function together with the Btn2 paralogue Cur1 and the Hsp40 Sis1 to redirect misfolded proteins to various sequestration sites[68]. Cur1, as noted in another study, appears to be missing the Btn2 CTD which we find to be the SUMOylated domain[13]. While there is no evidence of a role for Cur1 in DNA damage induced PQC, integrating Cur1 in future studies will be important. Additionally, Sis1 interacts with the NTD of Btn2 during heat shock to further recruit Hsp104 to INQ sites[13]. Since we noted no major change in Hsp104 localization in Btn2-6KR cells (Fig. 5d), we did not pursue Sis1 any further here. Nonetheless, determining whether heat shock coupled with proteasomal impairment results in Btn2 and Hsp42 SUMOylation, thereby affecting sequestration in a DNA damage independent manner will be an interesting future direction.

What can INQ tell us about nuclear PQC in response to DNA damage across species? SUMOylation of human proteins in the DNA damage response has been recognized for many years, for example in the regulation of BLM helicase localization to PML bodies[69]. Indeed, PML has been shown to function in nuclear PQC in a SUMO dependent manner along with the SUMO targeted ubiquitin ligase RNF4[70]. Bringing these studies together provides dramatic parallels with the yeast system we describe here. Moreover, just like the relationship between INQ and CytoQ frequency we observed, the nuclear SUMO dependent network described in humans also cross-talks with cytoplasmic stress granules[71]. While PML protein is not conserved, and PML bodies seem to serve multiple functions in humans, organizing principles related to PQC functions seen at INQ are clearly evident. Additionally, sequestration as a PQC response in humans has been discovered and elucidated at the aggresome (reviewed in ref. [72]). While cytoplasmic sequestration appears to be conserved, conservation of nuclear sequestration and compartmentalization of human misfolded proteins in an INQ-like structure remains to be discovered. Whether other subnuclear PQC compartments in human cells better align with INQ remains to be seen. Studies of humanized yeast suggest that at least human sHsps HspB2 and HspB3 have the potential to serve as sequestrases[73]. Moreover, HspB2 and B3 localize to the nucleus and can form phase separated droplets in human cells, supporting the potential for a conserved pathway of nuclear PQC by sHsps with sequestrase functions[74]. Future studies should help to further define key conserved features of nuclear PQC in response to DNA damaging stresses and the contributions of sHsps, and SUMO to this process.

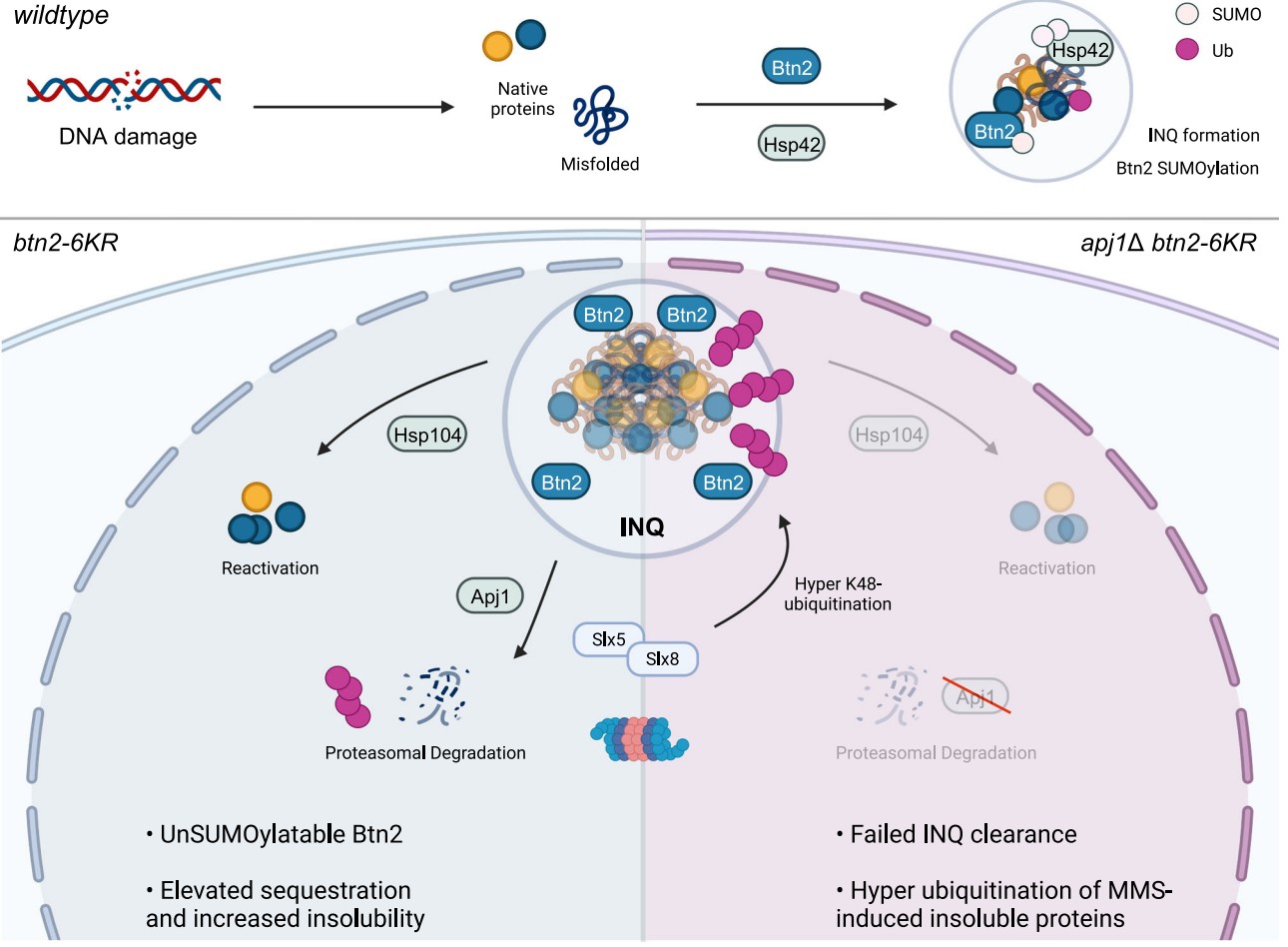

**Fig. 7 | Btn2 SUMOylation regulation of INQ dynamics.** See main text for additional details. Briefly, in WT cells (top), Btn2 SUMOylation likely occurs after INQ formation and helps to regulate normal ubiquitination and turnover of substrates during stress recovery. In cells with Btn2-6KR (bottom) INQ forms normally, but

Btn2 and substrates like Rpd3 shift more strongly to the insoluble fraction suggesting a potential change in state. Cells that further lack Apj1 (lower right) completely fail to clear INQ and exhibit hyper-accumulation of ubiquitinated proteins in the insoluble fraction. Created with BioRender.com.

## Methods

### Experimental model and subject details

Yeast (*Saccharomyces cerevisiae*) strains used in this study (s288c), database IDs, genotypes, primers, and plasmids used in this study are listed in the Supplementary Data 1. Except for the plasmids used in Fig. 4e–g, and Supplementary Fig. 3c–f, the remaining GFP, RFP and mCherry tagged strains mentioned are genomically integrated. All strains were grown under nutrient rich YPD medium[75] or synthetic medium lacking amino acids for auxotrophic selection unless otherwise indicated. Serial dilution assays and growth curve analyses were performed as described previously in refs. [15,76]. Briefly, cells with identical optical density (OD) were serially diluted tenfold and spotted on YPD plates using a 48-pin replica pinning manifold then incubated at the indicated temperatures for 72 h. Growth curves were analyzed in a Tecan M200 plate reader monitoring OD600 nm every 30 min for 48 h at 30 °C. For all MMS treatment conditions, log cells were exposed to 0.05% MMS (~99%; Sigma) in synthetic complete (SC) medium for 2 h unless otherwise noted. For growth curves and washouts, MMS was then washed out and replaced with fresh MMS-free medium. All other agents DNA damaging agents used were hydroxyurea (200 mM), hydrogen peroxide (2 mM), camptothecin (25 μM), and ethyl methanesulfonate (0.5%) for 2 h in SC medium before imaging. Plasmid constructs were created using the pRS316 plasmid backbone and the NotI and Kpn1 restriction sites. Btn2 CTD lysine-to-arginine mutants were created using gene blocks that were purchased from Integrated DNA Technologies. Constructs included necessary fragments to stitch

a full length Btn2 into pRS316 that were used for further experiments. Primers are listed in Supplementary Data 1.

### Live-cell imaging, image acquisition, analysis, and statistical methods

Imaging and subsequent analysis were performed as described previously[15]. Log-phase yeast were mounted on slides pre-treated with concanavalin A, in SC growth medium. For DAPI (4′,6-diamidino-2-phenylindole) staining, log-phase cells were fixed with ice-cold 70% ethanol on ice for 5 min and washed with $H_2O$. Fixed cells were then incubated with 50 ng/ml DAPI in phosphate-buffered saline for 15 min at room temperature before being mounted on pre-treated concanavalin A slides. Immobilized cells were imaged using an Objective HCX PL APO 1.40 NA oil immersion 100× objective on an inverted Leica DMi8 microscope with a motorized DIC (differential interference contrast) turret (for DIC imaging) and a filter cube set for FITC/TRITC (for GFP and RFP/mCherry fluorescence imaging). The images were captured at room temperature using a scientific complementary metal oxide semiconductor (sCMOS) camera (ORCA Flash 4.0 V2; Hamamatsu), collected using MetaMorph Premier acquisition software Version 7.8 and post processed [including gamma adjustments, counting of cells with/without foci, Z-stacking, projection of maximum intensity and foci intensity measurements] using ImageJ Version 5.3t (National Institutes of Health). For all microscopy experiments, the significance of the differences was determined using Prism version 8 or higher (Graphpad). For foci intensity measurements, regions of

interest were defined such that only the INQ focus per foci positive cell was used for further analysis. Intensities were analyzed for >100 INQ foci spanning three independent experiments. Samples were compared with one-way ANOVA; Prism performs $F$ tests for variance as part of this analysis. For comparisons of proportions, Fisher tests were used, and $P$ values were Holm–Bonferroni corrected in the event of multiple comparisons. Sample sizes and specific statistical details for each image analysis are listed in the figure legends.

## Western blotting and protein stability time course
For western blotting, whole-cell extracts of logarithmic phase cells were prepared using trichloroacetic acid (TCA) extraction and blotted with experiment-specific antibodies. For the stability time course, overnight cultures of the indicated strains were diluted to below an OD600 nm of 0.2 and allowed to progress into the logarithmic phase before collection. The cells were treated with or without cycloheximide (CHX) at a concentration of 200 µg/ml for the indicated times (0–120 min). Final samples of $2 \times 10^7$ cells were collected by centrifugation, and whole-cell extracts were prepared by TCA extraction and used for immunoblotting. All experiments were conducted in triplicates unless specified otherwise.

## Ni-NTA pulldown of SUMOylated proteins
Denaturing nickel bead pulldowns of His-SUMO conjugates were carried out as previously described[16]. In brief, strains of interest were transformed with a His-tagged SUMO plasmid (Smt3–His×7; Addgene #99538). Logarithmically growing cells were harvested at an OD600 nm for a total of $2 \times 10^9$ cells and collected by centrifugation (4000 rpm, 5 min at 4 °C). Harvested cells were washed and resuspended in 5 ml of pre-chilled water, lysed with 800 µl of 1.85 M NaOH containing 7.5% (v/v) 2-mercaptoethanol, followed by 20 min incubation on ice. Protein precipitation was carried out by adding 800 µl 55% TCA on ice for 20 min. Precipitates were pelleted by centrifugation (8000 g, 20 min, 4 °C) and resuspended in 1 ml Buffer A (6 M guanidine hydrochloride, 100 mM sodium phosphate and 10 mM Tris-HCl, pH 8.0). After incubation on a rotating block at room temperature for 1 h to solubilize the precipitate, the resulting solution was centrifuged (16,000 × g, 10 min, 4 °C), and the supernatant was transferred to tubes containing 60 µl precleared Ni-NTA agarose beads (Qiagen) in the presence of 0.05% Tween-20 and 15 mM imidazole. Samples were incubated overnight on a rotating block at room temperature. The following day, the beads were washed twice with Buffer A containing 0.05% Tween-20 and four times with Buffer C (8 M urea, 100 mM sodium phosphate, and 10 mM Tris-HCl, pH 6.3) containing 0.05% Tween-20. Beads were centrifuged (200 g, 15 s) and supernatant was completely removed. His–SUMO conjugates on the beads were eluted by adding 30 µl loading sample buffer (8 M urea, 5% SDS, 200 mM Tris-HCl, pH 6.8, 0.1 mM EDTA, 0.5% Bromophenol Blue and 15 mg/ml dithiothreitol) and heating at 70 °C for 10 min. Resultant protein extracts were subjected to standard western blotting and probed for SUMO and other proteins of interest using antibodies as described above.

## Fractionation of the insoluble pellet
Cellular fractionation was performed as described in refs. 58. In brief, logarithmic cells were collected after or before MMS treatment. Frozen cells were resuspended in a fractionation buffer (100 mM HEPES, 1% Triton X-100, 300 mM NaCl, protease inhibitor cocktail) and lysed by bead beating. Lysates were precleared at 100 g, 4 °C for 5 min. Supernatant (whole cell lysate) was recovered and fractionated at 16,000 g, at 4 °C for 10 min to separate into soluble (S) and insoluble (P) fractions. Equal amounts of each fraction were resolubilized in a loading sample buffer (8 M urea, 5% SDS, 200 mM Tris-HCl, pH 6.8, 0.1 mM EDTA, 0.5% Bromophenol Blue, and 15 mg/ml dithiothreitol), heated at 70 °C for 10 min and were loaded onto gels followed by immunoblotting.

## Filter trap assay for detection of insoluble protein species
A serial dilution of fractionated samples was subjected to vacuum filtration through a cellulose acetate membrane (0.2 µm, Sterlitech) using a Slot Blot apparatus as described in ref. 77. Insoluble species retained on the membrane were analyzed using anti-GFP. In parallel, whole cell lysates were also run on a PVDF Western blot to confirm expression levels. Experiments were performed in triplicates and intensities were measured using ImageLab Version 5.2.1.

## Statistics and reproducibility
Data collection and statistics were performed in Microsoft Excel and GraphPad Prism (Version 8 and higher). All microscopy experiments and filter trap assays were performed in three independent biological replicates unless specified. Nickel-His SUMO Pulldowns and fractionation assays were performed in three independent biological replicates unless specified. For example, pulldowns comparing Btn2-WT and Btn2-6KR SUMOylation were performed over four independent replicates. Microscopy data is represented in ±SEM and analyzed using Fisher's two-tailed $t$ test, and INQ intensities were compared using ordinary one-way ANOVA. $****p < 0.0001$, $***p < 0.0002$, $**p < 0.002$, $*p < 0.03$, ns, $p > 0.1$.

## Reporting summary
Further information on research design is available in the Nature Portfolio Reporting Summary linked to this article.

## Data availability
All data supporting the findings of this study are available within the paper and its Supplementary Information. Source data are provided as a Source Data file. Further information and requests for resources and reagents should be directed to and will be fulfilled by the corresponding authors upon request. Source data are provided with this paper.

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

## Acknowledgements

We thank Phil Hieter and Elizabeth Conibear for providing the C- and N-terminal GFP fusions of all the strains used in the paper; Helle Ulrich for the Smt3-Hisx7 plasmid, and members of the Stirling lab for technical assistance. P.C.S. is funded by a Natural Sciences and Engineering Research Council of Canada Discovery grant and Discovery Accelerator Supplement (RPGIN 2020-04360), and a Canadian Institutes of Health Research Project grant (MOP136982).

## Author contributions

Conceptualization: A.K., P.C.S.; Methodology: A.K., V.M.; Investigation: A.K.; Writing - A.K., P.C.S.; Visualization: A.K., P.C.S.; Supervision: P.C.S.; Project administration: P.C.S.; Funding acquisition: P.C.S.

## Competing interests

The authors declare no competing interests.
