## [Peer Review File · Nature Communications]

Dynamics of DNA damage-induced nuclear inclusions are regulated by SUMOylation of Btn2REVIEWER COMMENTS

Reviewer #1 (Remarks to the Author):

The authors show that the small heat shock protein sequesterases Btn2 is SUMO modified and linked to protein clearance. This further demonstrates the importance of post-translational modifications to protein sequestration in the nucleus. Further, they show that it is the SUMOylation of the chaperones and not the substrates that links SUMO to the INQ. They tie SUMOylation to the C-terminal domain of Btn2, but don't isolate the exact residue(s) that are modified. They link chaperone SUMOylation to DNA damage. They further show that Btn2 SUMOylation prevents further accumulation of proteins into the INQ but does not prevent INQ formation. They posit that Btn2 is SUMOylated to counteract overactive Slx5/8 and prevents the accumulation of K48-Ub chain modified proteins.

While of enormous potential significance to the field of protein homeostasis and disease, there are many critical issues that limit the significance and impact of the study. Several important papers aren't discussed in the manuscript including ones that have opposing findings, some of the points listed as novel have already been described in the literature, and a few of the experiments are not directly testing the question but instead going at it from a round-about way. These issues are described in detail below.

Critical Issues:

Sis1 and Cur1: sHsps Sis1 and Cur1 have been shown to sequester misfolded proteins to the nucleus, perinuclear region, and cytoplasm depending on the interacting partners including Btn2 and Hsp42. These proteins are never mentioned in the entire manuscript despite being essential to any discussion of sorting to specific inclusion locations. Critical references relevant to the Sis1-Cur1-Btn2-Hsp42 findings are also missing including Malinovska et al. 2012, Specht et al. 2011, Park et al. 2013, Jawed et al. 2023. Malinovska et al. shows that Btn2 works with Hsp42 to sort misfolded proteins to the periphery, Btn2-Sis1 to the perinuclear region, and Sis1-Cur1 into the nucleus. This contrasts with the finding of this manuscript that Btn2 alone can sequester misfolded proteins inside the nucleus, yet this discrepancy is never addressed.

Rpd3 as INQ marker: Figure 1 very nicely shows that Rpd3 can form cytoplasmic foci, yet it is used as an INQ marker throughout the entire manuscript. I am not confident that true nuclear inclusions can be differentiated from cytoplasmic, perinuclear inclusions using only Rpd3 as a marker, especially without a nuclear membrane stain to visualize the nuclear boundary. Lines 119-120 state that Rpd3 is in both INQ and CytoQ inclusions.

Major Issues:

Apj1: Apj1 has already been described as an Hsp40 chaperone with a role in SUMO-mediated protein degradation that promotes disaggregation of intranuclear protein inclusions (taken from Saccharomyces Genome Database), undercutting the novelty of Btn2 having the same role. This paper is cited (de Brave et al 2020, reference 49) but only to establish 2 independent pathways for INQ clearance and the similarity between the findings of this paper for Btn2 with de Brave et al. for Apj1 is not discussed.

Foci: Is there a set of criteria used to judge what is or isn't a focus? Some of the cells that supposedly contain foci look the same as others that are said to not contain foci. For instance, the ctd and ctd+clb deletions in figure 4f and the hsp42 deletion cells in supp fig 1a are said to not contain foci but the H2O2 and HU treated cells in figure 2a do contain foci. The cells in these images look incredibly similar, and it is then unclear how foci are being determined. Additionally, the quantitation graphs are labeled as "Cells with Rpd3-GFP foci (%)" and it's unclear if that is % of all cells or % of cells with Rpd3-GFP expression.

Figure 1a: How is nuclear vs cytoplasmic inclusion being determined? Based on the text and figure 1b, it appears that colocalization with Hta2 is being used as a nuclear marker. Miller et al. 2015 showed that inclusions can occur in the nucleoplasm that don't colocalize with DNA or histone staining. They demonstrate that a nuclear membrane marker must be used to accurately determine nuclear localization relative to the nuclear boundary.

Cellular fractionations: The fractionations were just separating soluble from insoluble proteins in whole cell lysates. In order to accurately determine if the insoluble material originated in the nucleus or cytoplasm, the nuclei must be isolated then lysed separately from the cytoplasm. The insoluble material is then extracted from nucleus and cytoplasm independently, ensuring that the insoluble material is from the INQ or CytoQ respectively.

Supp figure 1c: Ume1 and Rco1 clearly have bands in the pellet with MMS treatment but line 130 says that they remained soluble. Rco1 also has a band in the pellet without MMS treatment when you wouldn't expect insoluble material in the cell, but this is not discussed. Ume1 is in the figure but not the text and Snt2 is in the text but not the figure.

Figure 3c and lines 190-192: An increase in the % of cells with foci has been interpreted as increased INQ formation in the other figures, why would it indicate decreased clearance in this instance?

Figure 4b: If His-SUMO is being pulled down, how is there non-SUMOylated GFP-Btn2 and Hsp42-GFP? Then, if pulling down non-SUMOylated proteins, how do you know the laddering is SUMO and not ubiquitin or other modifications?

Supp fig 4d: Why not just do a filter-trap assay instead of complicated structure prediction. Filter trap will tell you if there is a difference in the solubility of Btn2.

Line 310: To determine if INQ is becoming more insoluble as a compartment, it would be more straight-forward to do a FRAP or FLIP experiment. Additionally, using photoconvertible fluorescent proteins would give an indication of diffusion in and out of the INQ to determine if it becomes more static or gel-like.

Fig 6b: GFP becomes hyperfluorescent in inclusions, therefore fluorescence intensity doesn't correlate to amount of Rpd3 in the INQ.

The full Western blots were not provided, nor was an antibody list with manufacturer and concentration used.

Minor Issues:

Lines 70-71: Sentence describes small, cytoplasmic foci but it is unclear if these are the cytoQ discussed in the paper or another type of cytoplasmic inclusion that is a precursor to the JUNQ. The JUNQ is introduced but never discussed, and cytoQ is discussed but never introduced.

Lines 71-73: No reference for this statement.

Line 75: Reference 11 leads to a BioRxiv paper that was published earlier this year in Nature Cell Biology PMID 37081164.

Line 89: Sentence states that Rpd3 is a new INQ marker, but line 103 states that Rpd3 has been shown to relocalize to the INQ (refs 14,16), and line 411 says that Rpd3 came up in a large-scale screen of candidate INQ proteins, but this study is the first direct test of Rpd3 as an INQ protein. This inconsistency is confusing.

Lines 97-98: States that SUMOylation is a previously unknown PTM of sHsps but Hsp42 is already known to get SUMOylated and the modification is annotated in the Saccharomyces Genome Database from Bhagwat NR, et al. (2021) PMID: 33502312. SUMOylation of Btn2 was previously unknown, but not all sHsps.

Line 105: MMS is not described. The acronym is defined, but what it is and why it is being used is not stated until lines 134-135.

Figure 1d: Y-axis is labeled "Cells with total Rpd3-GFP Foci (%)". I interpret that to mean that it is counting both nuclear and cytoplasmic foci, but the text (lines 112-113) makes a distinction between INQ and CytoQ. If this distinction is being made from images similar to the ones in supp fig 1a – it is not convincing that the btn2 deletion cells don't have nuclear foci.

Figure 2b and Supp fig 2a: The % of cells with INQ in Fig 2b adds up to 100% of cells for each mutation, but in Supp fig 2a the untreated cells are all around 30% of cells with INQ when they are labeled the same.

Figure 2b: Does the INQ get cleared after division leading to the no bud cells lacking an INQ when they have presumably already budded? Does the INQ specifically form during S-phase or it was just seen in small buds?

Line 185: What is the CH mutation in Mms21?

Figure 3b: What did the inclusions look like in these mutants? Were they larger or smaller, or a different number per cell?

Figure 3d: Did the ubiquitin and SUMO levels change in the soluble fraction too?
Figure 4: No antibody listed for Rfa1. C, e, and g don't list any antibodies for the blots. E and g don't give relative exposures like b and c.
Figure 4e and Supp figure 3c: Less steady state protein for the truncations (more prominent in Supp 3c). If only a fraction of the steady state protein is SUMOylated, it's likely that any SUMOylated truncation protein would be below the detection limit of the assay.
Figure 4e: Why not include a GFP-tagged CTD only to show that it is SUMOylated? Why rely on a loss of signal that could happen for any number of reasons?
Supp figure 3d: Why is the INQ forming in Btn2-wt without any stress? How is it more functional if it's forming something that shouldn't be there in the non-stressed condition?
Line 260: Neither figure 4g nor supp fig 3f are showing an Smt3 stained blot.
Fig 4g, supp fig 3f and g: none of the GFP-Btn2-SUMO blots are very convincing that 6KR has decreased SUMOylation. Supp 3g is maybe a little more convincing than the other 2, but none of them are really strong.
Supp fig 4a: Why wasn't the 0.05% MMS used here when it has been used the entire manuscript? It would be very useful to compare to the rest of the experiments.
Supp fig 4b: Very difficult to see with the colors used, especially the tan. The dotted lines are not defined.
Line 285: Why/how would CHX dissolve the INQ? Especially without an increase in soluble Btn2? If it's not being cleared and not being solubilized, where is it going?
Line 286: INQ formation is not blocked with CHX as there is insoluble material at T0, indicating INQ formation.
Line 381: No image of Slx8-GFP to show localization in the INQ. How is it independent of SUMOylation status when the 6KR has significantly more Slx8-GFP foci than WT?
Line 382: data discussed corresponds to Fig 6f, not e
Line 388: data discussed corresponds to Fig 6e, not f
Line 581: Is 5 min dehydration in EtOH sufficient fixation to retain puncta? Most papers use 4% PFA.
Lines 610-616: Doesn't state how the pellet is resolubilized for SDS-PAGE analyses.

Reviewer #2 (Remarks to the Author):

In this manuscript, the authors investigated the role of SUMOylation in nuclear protein quality control at INQ in response to DNA damage stress. They show that Rpd3 protein, a histone deacetylase in budding yeast, is sequestered at INQ foci when treating the cells with MMS or other DNA damaging agents. This Rpd3 sequestration under DNA replication stress is reduced upon deletion of the small heat shock protein Btn2 and importantly, partially regulated by Btn2 SUMOylation. The authors show for the first time that Btn2 is SUMOylated within the C-terminal domain in response to MMS treatment and that the non-SUMOylatable allele Btn2-6KR exaggerates Rpd3 sequestration suggesting a possible role for Btn2 SUMOylation in Rpd3 foci clearance at INQ. In addition, the role of Btn2 SUMOylation in Rpd3 INQ clearance is shown to be epistatic with the Hsp104 refolding pathway, and independent of the Apj1-mediated degradation pathway by negatively regulating K48 ubiquitin chain building in MMS-treated cells in the absence of Apj1.

This work describes a novel and interesting finding of a link between SUMOylation and protein quality control in the nucleus, and identifies the first SUMOylated protein in the INQ to be the sequester Btn2. However, the conclusion that Btn2 regulates nuclear protein quality control in DNA damage response is not yet fully supported as much of the findings pertain specifically to the Rpd3 protein; other INQ marker proteins do not appear to be similarly affected. Additionally, the differential effects of plus/minus SUMOylation are modest as presented, raising questions about the ultimate impact of the finding.

1. The authors statement that HSP42 deletion abrogated all inclusions is tempered by the observation that total Rpd3 levels appear to plummet in hsp42Δ cells. This result is not commented on in the text

and could obscure the relative fractionation profiles presented.

2. The authors show that *ubc9-ts* mutant reduces Rpd3 foci in MMS treatment at both 25°C and 30°C in Fig. 3a. The effects of this mutant at both temperatures are similar, raising the possibility that reduction of Rpd3 foci may not be due to disruption of SUMOylation function but due to something else (for example, reduction in Btn2 levels as observed in Fig. 4c). Can the authors demonstrate that *ubc9-ts* is hypomorphic at both 25°C and 30°C? Additionally, the reduction in Rpd3 foci in this panel is modest, at best – roughly a decrease from 38% to 25% cells displaying foci. As mentioned above, SUMOylation of Btn2 may be a modifier of INQ formation rather than a true regulator.

3. It has been shown that Stm3 localizes to INQ, and the authors show that the *smt3-3KR* mutant increases Rpd3 foci in MMS treatment. Does the *smt3-KR* mutant affect the solubility of Stm3? Because more Smt3 protein is detected in the pellet fraction of untreated sample (Fig. 3d). The increase in Rpd3 foci in the mutant could be due to increased insoluble Stm3 rather than polySUMOylation reduction, per se.

4. In supplementary figure 3g, two closely migrating bands are observed in the SUMO pulldown, making it difficult to discern if Btn2-6KR is still highly SUMOylated or not, despite the arrow pointing to the band the authors appear to like the best. Can the authors comment on the identity of the extra band or perhaps obtain a cleaner result?

5. The authors claim that the Btn2-6KR mutation does not affect solubility of Btn2 protein in silico using protein structure prediction software. While these tools may suggest a likelihood, it would be far preferable if the authors could simply produce purified Btn2 and Btn2-6KR from bacterial sources and validate this prediction. This is important, as the retention of the KR mutant protein in the pellet fraction (Fig. 5c) is either lack of SUMO or misfolding, two distinctly different rationales.

6. In Fig. 5d, the authors show that Btn2-6KR mutant highly increases Rpd3 foci. However, the effect of Btn2-6KR on other INQ markers is negligible, despite what seem to be aspirational significance numbers. For this reason, the claim in the title that SUMOylation of Btn2 “regulates nuclear protein quality control” seems an overreach. The authors are recommended to temper these claims with softer language, or identify additional substrates, even model ones, whose behavior mirrors that of Rpd3 more closely.

7. The result showing that Btn2-6KR partially ameliorates the cell growth defect of *fes1 hsp104* mutant cells is extremely interesting and in truth, the sole point of physiological relevance for the SUMOylation discovery. However, this experiment doesn't link at all to the original point of the story regarding the DNA damage response. Can the authors add a panel with these strains and varying concentrations of MMS in the plates? Heightened resistance to the drug would be very compelling.

Other comments:

8. The authors state in the methods that different statistical tests were used depending on the type of data – however, all figure legend indicate that only Fisher's test was used for every figure. Can the authors clarify?

9. In contrast to the localization quantitation and statistical validation, no such rigor is found in the protein work – SUMOylation, subcellular fractionation, etc. Assuming the experiments were repeated multiple times, the authors should strongly consider quantitation of band intensities to strengthen their conclusions.

10. The references for *ubc9-1 ts* mutant, DNA damaging agents, and *Mms21-CH* are missing.

11. Fig. 6e mentioned in the text is Fig. 6f, vice versa.

12. In Fig. S1C, the fractionation data of Ume1 is shown instead of Snt2 as mentioned in the text.

Reviewer #3 (Remarks to the Author):

In this study the authors investigated DNA damage-induced sequestration of Rpd3 (a histone deacetylase and a chromatin remodeler) to the intranuclear protein quality compartment (INQ), and the role of INQ-protein SUMOylation in this process, using yeast. Although Rpd3 localization to INQ has been previously identified in a large screen, this study investigates it in more detail. Following an earlier report of SUMO localization to the INQ, the authors show that mutants defective in protein SUMOylation have a defect in Rpd3 targeting to INQ. Next, the authors tested which INQ-proteins may be SUMOylated and found that a small chaperone Btn2 is SUMOylated upon DNA damage. To investigate the role of Btn2 SUMOylation in INQ, the authors used a Btn2-6KR mutant. The data show that, upon removal of the DNA-damaging agent, cells expressing mutant Btn2-6KR have a delay in dissolution of Rpd3 from INQ, especially in the *apj1* mutant background with impaired proteasomal degradation of INQ-components, indicating Btn2-SUMO function within the same epistasis group with Hsp104-dependent INQ-protein extraction.

Together the findings presented in the paper provide additional characterization of DNA damage induced Rpd3 sequestration to INQ, and raise an interesting possibility that DNA damage induced SUMOylation of a small chaperone is important for efficient protein dissolution from INQ upon stress removal.

I find the claims of DNA damage induced Rpd3 re-localization to INQ, in a process that is specific for Rpd3 subunit of different Rpd3-complexes and requires small chaperones Btn2 and Hsp42, well supported by the data, as well as DNA damage-induced SUMOylation of Btn2. The claims based on using Btn2-6KR mutant require additional clarification as pointed below.

Since the area of DNA replication stress is outside the scope of my expertise, I was unable to assess this part of the data fully.

The paper is written clearly.

Suggested improvements:

1. High molecular weight bands that correspond to SUMO-ylated Hsp42 in Fig. 4 b seem very weak compared to unmodified Hsp42, suggesting that only a very small fraction of Hsp42 is modified by SUMO. Could the authors provide a comment on this, modify the text accordingly or present additional data that corroborates substantial levels of Hsp42 SUMOylation.

2. In Western blot analyses of Ni-PD (Fig 4 g, Fig. S3-e) two bands are visible above non-modified Btn2, the lower one has been marked as corresponding to SUMO-ylated Btn2. The upper band is not visible in *ctdD*-mutant and empty vector control (Fig. 4 e), it is very faint in *KalIR* mutant (Fig. S3-e, lane 3), but is stronger in the case of Btn2-6KR mutant (Figure 4 g lane 2, and Fig. S3-g lane 2). Based on this, my concern is that Btn2-K6R mutant is still SUMOylated and, therefore, that the results using this mutant are difficult to interpret. Could the authors comment on what is the evidence for diminished Btn2-K6R mutant SUMOylation, i.e. what is the evidence that the upper band visible in Ni-PD of Btn2-K6R mutant is not a SUMOylated protein. For instance, does this band disappear in *ubc9ts* or SUMO-E3 ligase mutants?

3. Following on the previous comment, Fig. 5 d shows that more cells expressing Btn2-6KR mutant

display Rpd3-GFP INQ-foci, however, SUMO-E3 mutants show a decrease in Rpd3-GFP INQ-foci (Fig. 3 b). Could the authors comment on this results.

Minor comments:

4. In order to more easily understand the data showed in Fig. 2 b, I suggest moving the Figure S2 a and b to the main Fig. 2.

5. The result shown in Fig. 3a is difficult to interpret since percentage of cells with Rpd3-foci in *ubc9ts* mutant decreases in cells incubated both at 25 C and 30 C. Therefore the impact of the temperature itself interferes with the interpretation of the *ubc9ts* mutation. The data on the effect of impaired protein SUMOylation (E3 ligase mutants) is clear, so I suggest moving the figure with *ubc9ts* mutant to the Supplementary.

6. Paragraph lines 183-193: has the effect of SUMOylation mutants been tested for Btn2-GFP and/or other INQ-components? If this conclusion does not refer to INQ in general, I recommend that authors specify that the conclusion refers to Rpd3-INQ.

Author replies to reviews are in black normal text, reviewer comments are in *blue italics*. Reviewer summary statements are abridged but all specific comments are listed.

REVIEWER 1:

The authors show that the small heat shock protein sequesterases Btn2 is SUMO modified and linked to protein clearance. This further demonstrates the importance of post-translational modifications to protein sequestration in the nucleus. Further, they show that it is the SUMOylation of the chaperones and not the substrates that links SUMO to the INQ. They tie SUMOylation to the C-terminal domain of Btn2, but don't isolate the exact residue(s) that are modified. They link chaperone SUMOylation to DNA damage. They further show that Btn2 SUMOylation prevents further accumulation of proteins into the INQ but does not prevent INQ formation. They posit that Btn2 is SUMOylated to counteract overactive Slx5/8 and prevents the accumulation of K48-Ub chain modified proteins.

While of enormous potential significance to the field of protein homeostasis and disease, there are many critical issues that limit the significance and impact of the study. Several important papers aren't discussed in the manuscript including ones that have opposing findings, some of the points listed as novel have already been described in the literature, and a few of the experiments are not directly testing the question but instead going at it from a round-about way. These issues are described in detail below.

Critical Issues: Sis1 and Cur1: sHsps Sis1 and Cur1 have been shown to sequester misfolded proteins to the nucleus, perinuclear region, and cytoplasm depending on the interacting partners including Btn2 and Hsp42. These proteins are never mentioned in the entire manuscript despite being essential to any discussion of sorting to specific inclusion locations. Critical references relevant to the Sis1-Cur1-Btn2-Hsp42 findings are also missing including Malinovska et al. 2012, Specht et al. 2011, Park et al. 2013, Jawed et al. 2023. Malinovska et al. shows that Btn2 works with Hsp42 to sort misfolded proteins to the periphery, Btn2-Sis1 to the perinuclear region, and Sis1-Cur1 into the nucleus. This contrasts with the finding of this manuscript that Btn2 alone can sequester misfolded proteins inside the nucleus, yet this discrepancy is never addressed.

We apologize for this oversight. We are certainly aware of the Btn2 paralogue Cur1 and the Hsp40 protein Sis1. While there is good data on these factors in the context of prions, heat shock, and when the proteasome is disrupted, there is considerably less known about responses to DNA damaging stresses like MMS as used in our study. For example, the foci monitored in Malinovska are induced by heat, whereas Rpd3 does not form nuclear foci at high temperatures (**shown below** - Rpd3-GFP cells were subjected to heat shock at 42°C for 30 minutes similar to the Apj1-INQ den Brave paper protocol (PMID 32492414)).

Similarly, as noted by Ho et al. 2019 (PMID 31649258), Cur1 appears to be missing a C-terminal domain when compared to Btn2. Since this is also the SUMOylated region, we believe Cur1's role in sorting might be distinct from that of Btn2 SUMOylation. Regarding Sis1, it has been shown that Btn2 recruits Hsp104 to INQ sites through its interaction with Sis1 via its N-terminal domain (Ho et al., 2019 (PMID 31649258)). Since Hsp104 recruitment is not dramatically affected in Btn2-6KR cells (**Figure 5d**), we did not pursue Sis1 recruitment to INQ in the context of Btn2 SUMOylation, although it is an exciting future direction. Regardless, we agree with the reviewer and now put our findings in the context of literature on Sis1 and Cur1. To address this we have added a paragraph to the discussion which reads:

"It is important to note that our study investigates the role of SUMOylation at INQ under the context of DNA damage. Other studies have shown that stresses such as heat shock and proteasomal impairment also result in sequestration of proteins to INQ. Under these conditions, Btn2 and Hsp42 have been shown to function together with the Btn2 paralogue Cur1 and the Hsp40 Sis1 to redirect misfolded proteins to various sequestration sites. Cur1, as noted in another study, appears to be missing the Btn2 CTD which we find to be the SUMOylated domain. Additionally, Sis1 interacts with the NTD of Btn2 during heat shock to further recruit Hsp104 to INQ sites. Since we noted no major change in Hsp104 localization in Btn2-6KR cells (**Fig. 5d**), we did not pursue Sis1 any further. But determining whether heat shock coupled with proteasomal impairment results in Btn2 and Hsp42 SUMOylation, thereby affecting sequestration in a DNA damage independent manner will be an interesting future direction."

Rpd3 as INQ marker: Figure 1 very nicely shows that Rpd3 can form cytoplasmic foci, yet it is used as an INQ marker throughout the entire manuscript. I am not confident that true nuclear inclusions can be differentiated from cytoplasmic, perinuclear inclusions using only Rpd3 as a marker, especially without a nuclear membrane stain to visualize the nuclear boundary. Lines 119-120 state that Rpd3 is in both INQ and CytoQ inclusions.

Our exploration of Rpd3 as an INQ marker arose from high-throughput studies that identified Rpd3 as a candidate INQ resident (Gallina et al., 2015). It also co-localizes with Hos2 which has been used more often as an INQ marker. As rightly noted by the reviewer later in their comments, Miller et al. 2015 showcased the importance of using a nuclear membrane marker to differentiate between nuclear and cytoplasmic sequestration sites. To further support the nuclear localization of Rpd3 inclusions we analyzed Rpd3-GFP foci in cells expressing the nuclear periphery marker Nic96-mCherry, a component of nuclear pores. As can be seen in revised **Figure 1c**, exposure to MMS results in the sequestration of Rpd3 to intranuclear sites

within the boundary of the Nic96-mCherry signal. These foci co-localize with diffused Rpd3 nuclear staining, allowing us to confidently differentiate between nuclear and cytoplasmic Rpd3 foci. While Rpd3's normal localization to the nucleus along with our imaging and genetic data support Rpd3 as a nuclear inclusion marker, we agree that peripheral cytoplasmic Rpd3 foci are evident.

Major Issues:

Apj1: Apj1 has already been described as an Hsp40 chaperone with a role in SUMO-mediated protein degradation that promotes disaggregation of intranuclear protein inclusions (taken from Saccharomyces Genome Database), undercutting the novelty of Btn2 having the same role. This paper is cited (de Brave et al 2020, reference 49) but only to establish 2 independent pathways for INQ clearance and the similarity between the findings of this paper for Btn2 with de Brave et al. for Apj1 is not discussed.

We certainly did not intend to under-appreciate the importance of den Brave et al., 2020. Apj1 is clearly a specific regulator of INQ protein degradation with still unclear connections to SUMO recognition. Presumably as an Hsp40 Apj1 works with Hsp70 and the proteasome to disaggregate and degrade INQ substrates. Our work builds on this but the role of Apj1 is not the same as Btn2. Btn2 is a small heat shock protein that promotes sequestration of proteins at INQ, with a new role for SUMO regulation of this process described here. In fact, we think that our linkage of SUMO modification to an INQ chaperone provides compelling new directions for research on Apj1 that will be the basis of other work. We apologize for not making the distinctions clearer and have added to the discussion on Apj1 and Btn2 as follows:

“Our data on INQ clearance and abundance following washout of MMS supports the view that Apj1 and Btn2-SUMO work in parallel to promote turnover of proteins from INQ. Additionally, loss of Btn2 SUMOylation might impair the ubiquitin proteasome system (UPS) mediated degradation of specific substrates, since published work indicates that impaired UPS mutants also rescue the *fes1Δhsp104Δ* growth defect similar to Btn2-6KR. Indeed, we observed that cells with non-SUMOylatable Btn2 accumulate K48-linked ubiquitin modified proteins in the insoluble fraction when Apj1 was absent.”

*Foci: Is there a set of criteria used to judge what is or isn't a focus? Some of the cells that supposedly contain foci look the same as others that are said to not contain foci. For instance, the *ctd* and *ctd+cld* deletions in figure 4f and the *hsp42* deletion cells in supp fig 1a are said to not contain foci but the H2O2 and HU treated cells in figure 2a do contain foci. The cells in these images look incredibly similar, and it is then unclear how foci are being determined. Additionally, the quantitation graphs are labeled as “Cells with Rpd3-GFP foci (%)” and it's unclear if that is % of all cells or % of cells with Rpd3-GFP expression.*

Regarding the first point about foci scoring: We apologize for including representative images that do not highlight foci clearly, this was an error on our part and by no means suggests any challenges in calling foci. Foci are evident as regions of fluorescent intensity at least 2-fold greater than the surrounding nucleoplasmic staining that are demarcated by a round shape. In practice these are easy to score and we have replaced the images in **Figure 2a** to reflect this.

For the second point, all cells express Rpd3-GFP as the construct it is integrated into the genome. Cells with Rpd3-GFP foci (%) therefore means the proportion of all cells. We have clarified this point in the methods.

Figure 1a: How is nuclear vs cytoplasmic inclusion being determined? Based on the text and figure 1b, it appears that colocalization with Hta2 is being used as a nuclear marker. Miller et al. 2015 showed that inclusions can occur in the nucleoplasm that don't colocalize with DNA or histone staining. They demonstrate that a nuclear membrane marker must be used to accurately determine nuclear localization relative to the nuclear boundary.

We have now added additional imaging with Nic96-mCherry, a nuclear periphery marker used by Miller et al., 2015. This can be found in **Figure 1c** and confirms that Rpd3 forms nuclear foci. For most figures, nuclear foci are GFP inclusions that are adjacent to the nuclear GFP signal of Rpd3, whereas peripheral/cytoplasmic foci are separated from the main nuclear GFP signal for Rpd3. In our hands scoring with Hta2 (histone), Nic96 (nuclear membrane) or simply the Rpd3-GFP intense region marking the nucleus give similar results (**see below**).

Cellular fractionations: The fractionations were just separating soluble from insoluble proteins in whole cell lysates. In order to accurately determine if the insoluble material originated in the nucleus or cytoplasm, the nuclei must be isolated then lysed separately from the cytoplasm. The insoluble material is then extracted from nucleus and cytoplasm independently, ensuring that the insoluble material is from the INQ or CytoQ respectively.

This is an interesting point and unfortunately we have not yet refined a robust INQ purification strategy. Indeed, we are working on this as part of ongoing work outside of this study. In this study, fractionation is intended to show the total segregation of reporter proteins to the insoluble pellet irrespective of their localization. We have tried to be clear in our intent and interpretation throughout the text. For example in Figure 1f, Rpd3 partitions to the insoluble pellet in response

to MMS while its partners in the Rpd3S and L complexes do not (Supplementary Figure 1c), and this behavior is regulated by Btn2 and Hsp42. This is an important insight for our study regardless of which PQC compartment is being queried. Later the effects of Btn2-6KR on both Btn2 and Rpd3 solubility are assessed, but again it is the bulk behaviour that we are interested in. The sequestration of proteins into insoluble aggregates following DNA damaging stress is measured by these fractionation experiments.

Supp figure 1c: Ume1 and Rco1 clearly have bands in the pellet with MMS treatment but line 130 says that they remained soluble. Rco1 also has a band in the pellet without MMS treatment when you wouldn't expect insoluble material in the cell, but this is not discussed. Ume1 is in the figure but not the text and Snt2 is in the text but not the figure.

The reviewer is correct that we oversimplified this result. There is indeed a small amount of Ume1 or Rco1 in the pellet fractions after or before MMS treatment. In untreated cells, this is likely due to the conditions of the fractionation enabling chromatin-associated proteins to be precipitated at some level. Indeed, Rpd3 partitions weakly to the pellet in untreated cells in Figure 1. For Rco1, this pellet fraction does not increase under MMS treatment, suggesting no further partitioning to the pellet consistent with the imaging data finding no Rco1-GFP foci. For Ume1-GFP there does appear to be a slight increase in the pellet after MMS treatment. It is possible that this represents a real shift of another Rpd3L complex subunit to INQ, however, we do not support this view for two reasons. First, there is no evidence of Ume1-GFP forming foci in MMS. Second, the magnitude of the shift to the pellet in MMS is very small compared to Rpd3 which, in Figure 1, shows a very strong band evident in the pellet fraction.

We have amended the text to account for this (Page 3), but still conclude that Rpd3 is somewhat unique among other complex subunits and certainly the only subunit capable of serving as a robust INQ marker.

Figure 3c and lines 190-192: An increase in the % of cells with foci has been interpreted as increased INQ formation in the other figures, why would it indicate decreased clearance in this instance?

Since mutations in *UBC9* (the E2 for all SUMO in yeast) dramatically reduced INQ formation it suggested that SUMO is required for stable INQ. Mutations in *SMT3* (SUMO) itself that prevent poly-SUMO chains actually seem to promote INQ formation. Since mono-SUMOylation must precede poly-SUMOylation we are proposing a model that reconciles these findings (i.e. that SUMO in general is important to form INQ, but poly-SUMO actually antagonizes INQ). Indeed, we also note polySUMOylation for Btn2 (**Figure 6d and Supplementary Figure 3g**) confirming the poly-chain modification of INQ residents. Nevertheless, decreased clearance or more rapid assembly are the two possible mechanisms that would lead to excess INQ foci counts and we have now revised the text to reflect this possibility on page 5.

Figure 4b: If His-SUMO is being pulled down, how is there non-SUMOylated GFP-Btn2 and Hsp42-GFP? Then, if pulling down non-SUMOylated proteins, how do you know the laddering is SUMO and not ubiquitin or other modifications?

This is an interesting question, one for which we do not have the answer to at the moment. We note that for substrates such as Rpd3, Hos2, Cdc40, and Hsh155, no unmodified bands are present in the His-SUMO pulldowns. However, blots for chaperones such as Hsp104, Btn2, and Hsp42 contain non-SUMOylated bands. This could be due to an intrinsic property of chaperones, that causes them to be sticky resulting in non-SUMOylated forms of the protein being pulled down. While this could answer the question, other His-SUMO studies in both yeast and humans (PMID 34348159 Figures 5-7, PMID 31519521 Figure 3a, 4a, PMID 29549242 Figure 3b) have results wherein unmodified forms of a SUMOylated protein are visible suggesting that these could be a by-product of the method itself. Thus, unmodified bands sticking to the beads in this method are not unique and can be controlled for as we have done.

We used the *ubc9-1* temperature-sensitive allele to address the second question. There is a clear loss of the modified band in this background indicating to us that the modification is SUMOylation. Later in the study, we note that loss of *SLX5* results in stabilization of Btn2 SUMOylation and visible polySUMOylation of Btn2 (**Figure 6d**). This is in line with the function of Slx5 as a SUMO-targeting Ubiquitin ligase, further supporting that the modified Btn2 band being pulled down is true SUMOylation.

Supp fig 4d: Why not just do a filter-trap assay instead of complicated structure prediction. Filter trap will tell you if there is a difference in the solubility of Btn2.

We thank the reviewer for this suggestion. We ran our fractionated cells through a cellulose-acetate membrane to confirm they were in an insoluble aggregated state (**Figure 5c**). We loaded 0ug, 10ug, and 20ug of whole cell lysates and the pellet fraction for GFP-Btn2-WT and GFP-Btn2-6KR untreated cells and probed for GFP-Btn2. As you can see in the revised Figure, these experiments confirm that the protein isolated in the pellet fraction are indeed lysis-buffer resistant insoluble aggregates that interact with the cellulose-acetate membrane, and that Btn2-6KR is more insoluble than Btn2-WT.

Line 310: To determine if INQ is becoming more insoluble as a compartment, it would be more straight-forward to do a FRAP or FLIP experiment. Additionally, using photoconvertible fluorescent proteins would give an indication of diffusion in and out of the INQ to determine if it becomes more static or gel-like.

We agree that additional studies into the state and phase of INQ sequestration compartments would be a nice extension of the work. We regret that we are not set up to do these experiments in the time frame of this revision but point out that a detailed study of INQs biophysical characteristics would be beyond the scope of this work and a great topic for future research.

Kumar et al.

Fig 6b: GFP becomes hyperfluorescent in inclusions, therefore fluorescence intensity doesn't correlate to amount of Rpd3 in the INQ.

The intensities scored in **Figure 6b** are for regions of interest that only encompass the INQ focus itself (across a number of single cells). Therefore, all measurements will be of GFP embedded in INQ inclusions which we believe should allow us to compare across strains regardless of fluorescence emission changes associated with the state of the GFP fusion protein. Whether due to more molecules, or a state change, by focusing on GFP intensity within INQ foci, this data shows us how Btn2-6KR mutation synergizes with the loss of Apj1 to change the character of INQ. In other words, since we are only reporting the intensity of foci fluorescence the changes are meaningful. We did not intend to indicate a specific change in the number of molecules, simply that the fluorescence changes within foci formed in different genetic backgrounds match our other data. We have tried to clarify this in the methods. We could remove this data but think it is legitimate and supportive as presented.

The full Western blots were not provided, nor was an antibody list with manufacturer and concentration used.

We have added the uncropped western blots to the Source Data file along with the antibody list in both the Supplementary Data File and the reporting summary.

Minor Issues:

Lines 70-71: Sentence describes small, cytoplasmic foci but it is unclear if these are the cytoQ discussed in the paper or another type of cytoplasmic inclusion that is a precursor to the JUNQ. The JUNQ is introduced but never discussed, and cytoQ is discussed but never introduced.

We have amended our introduction of JUNQ and CytoQ on page 3: "While we can easily distinguish between nuclear and cytoplasmic foci, further distinction between the cytoplasmic foci (JUNQ vs IPOD) is difficult. To avoid miscalling, henceforth all cytoplasmic foci will be grouped and referred to as CytoQ inclusions.". While previous studies have used aggregation prone reporters to define these compartments, we are using endogenous proteins, expressed from their genomic loci. This makes unequivocal claims about whether peripheral foci are CytoQ or another compartment difficult to support. We have amended our language to reflect this.

Lines 71-73: No reference for this statement.

We have added the required references.

Line 75: Reference 11 leads to a BioRxiv paper that was published earlier this year in Nature Cell Biology PMID 37081164.

We have revised this to be the correct reference.

Kumar et al.

Line 89: Sentence states that Rpd3 is a new INQ marker, but line 103 states that Rpd3 has been shown to relocate to the INQ (refs 14,16), and line 411 says that Rpd3 came up in a large-scale screen of candidate INQ proteins, but this study is the first direct test of Rpd3 as an INQ protein. This inconsistency is confusing.

Sorry for the confusion. We have corrected this. Rpd3 was found in a high-throughput screen for INQ proteins, and we directly tested it for the first time here.

Lines 97-98: States that SUMOylation is a previously unknown PTM of sHsps but Hsp42 is already known to get SUMOylated and the modification is annotated in the Saccharomyces Genome Database from Bhagwat NR, et al. (2021) PMID: 33502312. SUMOylation of Btn2 was previously unknown, but not all sHsps.

We thank the reviewer for their comment. We were unaware of the Hsp42 annotation on SGD and have reworded the text to describe the context-specific SUMOylation of sHsps instead.

Line 105: MMS is not described. The acronym is defined, but what it is and why it is being used is not stated until lines 134-135.

We have revised the manuscript to introduce MMS and its mechanism of action at the beginning of the results section.

Figure 1d: Y-axis is labeled "Cells with total Rpd3-GFP Foci (%)". I interpret that to mean that it is counting both nuclear and cytoplasmic foci, but the text (lines 112-113) makes a distinction between INQ and CytoQ. If this distinction is being made from images similar to the ones in supp fig 1a – it is not convincing that the btn2 deletion cells don't have nuclear foci.

We have revised the graph to include both INQ and CytoQ levels.

Figure 2b and Supp fig 2a: The % of cells with INQ in Fig 2b adds up to 100% of cells for each mutation, but in Supp fig 2a the untreated cells are all around 30% of cells with INQ when they are labeled the same.

We apologize for the confusion. The untreated cells in **Supplementary Figure 2a** have around 30% of cells with spontaneous INQ as noted. We focused on this 30%, and split them into no bud, small bud and large bud cells as shown in **Figure 2b** thus resulting in a total of 100%. To make this clear, we have moved **Supplementary Figure 2a** to the main Fig. 2 ahead of the budding index.

Figure 2b: Does the INQ get cleared after division leading to the no bud cells lacking an INQ when they have presumably already budded? Does the INQ specifically form during S-phase or it was just seen in small buds?

For this figure, we used the budding characteristics of the yeast cells as a proxy for their cell-cycle phase. This process of performing a budding index has been an established protocol for successfully calling small-budded cells to be in S-phase. We note that cells that have small buds (S-phase) and large buds (G2/M) contain INQ, but cells with no buds (G1) have significantly lower levels of INQ. Since this experiment was done in untreated cells with mutations in DNA repair checkpoint kinases, we concluded that the replication stress in S-phase acts as a signal for sequestration of Rpd3 to INQ. While we did not perform a timelapse to monitor the clearance of Rpd3 INQ after G2/M in single cells, the relatively low amount of G1 cells that harbor INQ suggests that INQ may be cleared after division.

Line 185: What is the CH mutation in Mms21?

Mms21-CH contains two substitutions of C200A and H202A in the *MMS21* catalytic domain that inactivate its SUMO E3 ligase activity. We have added a reference for this mutation in the text.

Figure 3b: What did the inclusions look like in these mutants? Were they larger or smaller, or a different number per cell?

As we suggest in the paper, SUMOylation might be a sorting signal between INQ and JUNQ. Indeed, we see increased CytoQ formation along with a reduction in INQ formation (both inclusions look similar to WT inclusions). However, assessing the role of SUMOylation in the context of nucleocytoplasmic shuttling will require more biochemical experiments, controlled localization of substrates to either the nucleus or the cytoplasm, and a deeper analysis in general. For that reason, we did not delve deeper into this question since we feel it answers a larger question that is out of the scope of the current study.

Figure 3d: Did the ubiquitin and SUMO levels change in the soluble fraction too?

To address this we have now run the whole cell, soluble and insoluble fractions for all samples (WT, Smt3-3KR +/- MMS), and probed for ubiquitin and SUMO. As noted in other studies (PMID: 22285753, PMID 23122649), DNA damage induces a wave of SUMOylation and ubiquitination and the same is evident in the whole cell and soluble fractions upon MMS treatment. Indeed, ubiquitin fractions were lower in whole cell extracts of Smt3-3KR MMS-treated cells indicative of the role of polySUMOylation in ubiquitinating proteins. Surprisingly, the insoluble pellet fractions did not contain any free SUMO or ubiquitin. Instead, only high molecular weight SUMOylated and ubiquitinated conjugates partitioned to the inclusion-containing pellet fractions.

Figure 4: No antibody listed for Rfa1. C, e, and g don't list any antibodies for the blots. E and g don't give relative exposures like b and c.

We have added a full antibody list. Regarding the relative exposures, we are including the high exposure below but since it does not change the result nor our interpretation, we have excluded it from the main figure for now.

Fig. 4e

Fig. 4g

Figure 4e and Supp figure 3c: Less steady state protein for the truncations (more prominent in Supp 3c). If only a fraction of the steady state protein is SUMOylated, it's likely that any SUMOylated truncation protein would be below the detection limit of the assay.

We agree with the reviewer. Since truncations like *ctdΔ* and *ctd+ctdΔ* might be less stable, loss of SUMOylation might not be indicative of a true result but rather could be due to lowered stability of these constructs. Therefore, to confirm SUMOylation of Btn2 occurs at its C-terminus, we created the full-length GFP-Btn2-CTD-KallR construct which confirmed the C-terminus as the SUMOylated domain. Additionally, the 6KR mutation has significantly reduced SUMO. It is possible that low level constitutive SUMOylation also occurs elsewhere in Btn2 which we acknowledge in the text.

Figure 4e: Why not include a GFP-tagged CTD only to show that it is SUMOylated? Why rely on a loss of signal that could happen for any number of reasons?

We thank the reviewer for this suggestion. The CTD of Btn2 is highly disordered which was a point of concern for interpreting data on whether a GFP-CTD would act like a Btn2-CTD. Also this construct has been created by Ho et al. 2019 (PMID 31649258) and while it doesn't rescue INQ formation, in line with our results, immunoprecipitation experiments with this construct have increased laddering making it difficult to assess its stability. Indeed, a loss of SUMOylation signal for *ctdΔ* (C-terminal domain deleted) GFP-Btn2 cells could happen for several reasons. Which is why to ensure this loss of signal is a true result, we created a full-length Btn2-CTD-KallR construct in **Supplementary Figure 3e,f** instead.

Supp figure 3d: Why is the INQ forming in Btn2-wt without any stress? How is it more functional if it's forming something that shouldn't be there in the non-stressed condition?

We thank the reviewer for the opportunity to clarify. The presence of Btn2 GFP foci in some cells is consistent with INQ localization being a normal part of cell division and marking a site where protein turnover can take place. While particular endogenous substrates like Rpd3 may not accumulate to high enough levels without stress to mark INQ, Btn2 itself seems to be important for the formation of INQ structures and may serve as a good marker for basal levels of nuclear protein quality control. High throughput studies of N- and C-terminal GFP tagged yeast proteins (<https://shmoo.weizmann.ac.il/elevy/YeastRGB/HTML/YeastRGB.html>) support the view that GFP-Btn2 (N) is brighter and has less noise than Btn2-GFP (C) which is why we primarily use N-terminal GFP-Btn2 here and believe that it may be more reflective of native Btn2 than the C-terminal fusion.

It is important to note that Btn2 loss of function (i.e. through deletion or truncation) blocks INQ formation, so Btn2 localizing to INQ in unstressed is not consistent with the GFP tag causing loss of function. In addition, the formation of spontaneous foci at low levels is similar to other nuclear stress response proteins like Rad52, which localizes to DNA repair foci in ~10% of normal S-phases.

Line 260: Neither figure 4g nor supp fig 3f are showing an Smt3 stained blot.

Our rationale behind not including a Smt3 stained blot in these two figures is the same rationale for all the other SUMO NiPD pulldowns. We use Rfa SUMOylation as a positive control for both DNA damage and a successful pulldown. Since Rfa/RPA gets SUMOylated upon DNA damage in both **Figure 4g** and **Supplementary Figure 3f**, it is indicative of both a successful pulldown and MMS treatment. Indeed, looking for a conjugated substrate-SUMO species like RPA is a better control than free SMT3-His.

Fig 4g, supp fig 3f and g: none of the GFP-Btn2-SUMO blots are very convincing that 6KR has decreased SUMOylation. Supp 3g is maybe a little more convincing than the other 2, but none of them are really strong.

It is true that GFP-Btn2-6KR has a high background making it difficult to assess a clear loss of SUMOylation. To address this, we have repeated GFP-Btn2 NiPD SUMO pulldowns in Btn2-WT/6KR backgrounds to attain a cleaner result and also in a *ubc9-1* background to test for the slower migrating band above Btn2 SUMO (see below).

Repeating this experiment in *ubc9-1* cells revealed a complete loss of Btn2 SUMOylation in Btn2-WT cells indicating that Btn2-6KR might still be SUMOylated at a different site (**Panel a**). However, quantification of Btn2 SUMOylation in both Btn2-WT and Btn2-6KR cells over multiple experiments revealed a 2.7-fold reduction of Btn2 SUMOylation in Btn2-6KR cells (**Panel b**). Upon closer inspection of **Figure 4b (Panel c**, higher exposure) and repeating experiments in untreated conditions, we do find that Btn2 is SUMOylated even in untreated conditions (**Panel d**). While this might suggest that Btn2 SUMOylation increases upon MMS treatment due to increase in protein levels, we see a significant reduction in Btn2 SUMOylation in Btn2-6KR cells, back to low basal levels without any change in Btn2 protein level under MMS indicating that we do in fact abolish the DNA damage induced SUMOylation in the Btn2-6KR mutant.

Therefore, while Btn2-6KR might still be SUMOylated at low basal levels, we believe DNA damage induced SUMOylation is significantly reduced in Btn2-6KR cells. We have reworded the paper to reflect the same.

We would also like to clarify that **Supplementary Figure 3f** is comparing Btn2 SUMOylation status in Btn2-WT, -KallR, and -2KR cells only.

Supp fig 4a: Why wasn't the 0.05% MMS used here when it has been used the entire manuscript? It would be very useful to compare to the rest of the experiments.

We have now revised **Supplementary Figure 4a** and included 0.025% and 0.05% MMS concentrations as requested. We did not use these concentrations initially, since plating yeast on 0.05% MMS plates is highly toxic as compared to growing them in the presence of 0.05% MMS in liquid media for 2 hours. Regardless, even at these high concentrations of MMS, we do not notice any growth differences between Btn2-WT and Btn2-6KR.

Supp fig 4b: Very difficult to see with the colors used, especially the tan. The dotted lines are not defined.

The figure has been revised.

Line 285: Why/how would CHX dissolve the INQ? Especially without an increase in soluble Btn2? If it's not being cleared and not being solubilized, where is it going?

The observation about CHX treatment removing aggregate foci have been made by multiple other groups (PMID 25417105). One possible explanation is that CHX treatment, by stopping translation, alleviates the need for co-translational folding at ribosomes, dramatically increasing the concentration of free chaperones and increasing cellular refolding capacity. This could explain why CHX treatment results in INQ dissipation/dissolves INQ. However, this hypothesis has not been thoroughly tested and is not relevant to the manuscript and therefore we have removed this statement.

Line 286: INQ formation is not blocked with CHX as there is insoluble material at T0, indicating INQ formation.

Sorry for the confusion. In **Figure 5b**, we treated GFP-Btn2 cells with CHX and started collecting from T0. And the reviewer is correct, there is insoluble material at this time point indicating INQ formation. What we wanted to convey in Line 286 was pertaining to experiments conducted in other studies wherein a combined treatment of MMS and CHX abolishes INQ formation. We saw this in our study of another INQ protein Hsh155 (PMID 28978642) and based on the result from **Figure 5b**, we believe it could be explained by the removal of Btn2 from the insoluble fraction upon CHX treatment. Since this point is speculative and does not add to the experiment, we have removed this statement.

Line 381: No image of Slx8-GFP to show localization in the INQ. How is it independent of SUMOylation status when the 6KR has significantly more Slx8-GFP foci than WT?

We thank the reviewer for pointing this out. We intended to state that Slx8 is sequestered to INQ even if Btn2 cannot be SUMOylated. Such that, Btn2 SUMOylation is not the recruiting signal for the Slx5/8 complex. We have reworded that statement to avoid any confusion. Additionally, we have now included images for this experiment in **Supplementary Figure 5d**.

Line 382: data discussed corresponds to Fig 6f, not e

This mistake has been corrected.

Line 388: data discussed corresponds to Fig 6e, not f

This mistake has been corrected.

Line 581: Is 5 min dehydration in EtOH sufficient fixation to retain puncta? Most papers use 4% PFA.

While 4% PFA is widely used for fixation, a 5 minute treatment in 70% ethanol also seems sufficient to retain INQ and other cytoplasmic sequestration sites. The protocol we followed has been adopted from two other studies - Ho et al. 2019 (PMID 31649258) and Shrivastava et al. 2022 (PMID 36069810) from Bernd Bukau's group at ZMBH, Heidelberg - that focused on the role of the cytoprotective role of Btn2 and sequestration to INQ.

Lines 610-616: Doesn't state how the pellet is resolubilized for SDS-PAGE analyses.

The whole cell lysates, soluble and insoluble fractions were all resolubilized in a sample loading buffer (8 M urea, 5% SDS, 200 mM Tris-HCl, pH 6.8, 0.1 mM EDTA, 0.5% Bromophenol Blue and 15 mg/ml dithiothreitol). We have amended the text to reflect this additional step.

REVIEWER 2

In this manuscript, the authors investigated the role of SUMOylation in nuclear protein quality control at INQ in response to DNA damage stress. They show that Rpd3 protein, a histone deacetylase in budding yeast, is sequestered at INQ foci when treating the cells with MMS or other DNA damaging agents. This Rpd3 sequestration under DNA replication stress is reduced upon deletion of the small heat shock protein Btn2 and importantly, partially regulated by Btn2 SUMOylation. The authors show for the first time that Btn2 is SUMOylated within the C-terminal domain in response to MMS treatment and that the non-SUMOylatable allele Btn2-6KR exaggerates Rpd3 sequestration suggesting a possible role for Btn2 SUMOylation in Rpd3 foci clearance at INQ. In addition, the role of Btn2 SUMOylation in Rpd3 INQ clearance is shown to be epistatic with the Hsp104 refolding pathway, and independent of the Apj1-mediated degradation pathway by negatively regulating K48 ubiquitin chain building in MMS-treated cells in the absence of Apj1.

This work describes a novel and interesting finding of a link between SUMOylation and protein quality control in the nucleus, and identifies the first SUMOylated protein in the INQ to be the sequestrase Btn2. However, the conclusion that Btn2 regulates nuclear protein quality control in DNA damage response is not yet fully supported as much of the findings pertain specifically to the Rpd3 protein; other INQ marker proteins do not appear to be similarly affected. Additionally, the differential effects of plus/minus SUMOylation are modest as presented, raising questions about the ultimate impact of the finding.

1. The authors statement that HSP42 deletion abrogated all inclusions is tempered by the observation that total Rpd3 levels appear to plummet in hsp42Δ cells. This result is not commented on in the text and could obscure the relative fractionation profiles presented.

We thank the reviewer for pointing this out, as we actually did not pay much attention to the effects of Hsp42 on expression. Hsp42 appears to be essential for foci formation for many substrate proteins in the literature, not just Rpd3. We cite several such articles for various substrates (Hsh155 - PMID 28978642, Cmr1 - PMID 25817432, Cdc48 - PMID 33172985). We also use this as our rationale for pursuing work on Btn2, rather than Hsp42. The penetrant and pervasive effects of Hsp42 on cellular aggregation phenomenon throughout the cell make phenotypes more difficult to interpret. This is highlighted by the effects of Hsp42 on Rpd3 expression.

2. The authors show that ubc9-ts mutant reduces Rpd3 foci in MMS treatment at both 25oC and 30oC in Fig. 3a. The effects of this mutant at both temperatures are similar, raising the possibility that reduction of Rpd3 foci may not be due to disruption of SUMOylation function but due to something else (for example, reduction in Btn2 levels as observed in Fig. 4c). Can the authors demonstrate that ubc9-ts is hypomorphic at both 25°C and 30°C? Additionally, the reduction in Rpd3 foci in this panel is modest, at best – roughly a decrease from 38% to 25% cells displaying foci. As mentioned above, SUMOylation of Btn2 may be a modifier of INQ formation rather than a true regulator.

We have conducted additional experiments at 25°C and show that the *ubc9-1* allele is hypomorphic at both 25°C and 30°C (new **Figure 4c**) as seen by the loss of Btn2 SUMOylation at both temperatures. We believe this will help interpret the results of **Figure 3a** and that the reduction of Rpd3 INQ foci at both temperatures is due to perturbation of SUMOylation.

We are unclear on the distinction between Btn2 as a modifier of INQ versus a true regulator. Btn2 is clearly an INQ regulator based on the literature, Btn2-SUMOylation may be an INQ modifier.

3. It has been shown that Smt3 localizes to INQ, and the authors show that the smt3-3KR mutant increases Rpd3 foci in MMS treatment. Does the smt3-KR mutant affect the solubility of Smt3? Because more Smt3 protein is detected in the pellet fraction of untreated sample (Fig. 3d). The increase in Rpd3 foci in the mutant could be due to increased insoluble Smt3 rather than polySUMOylation reduction, per se.

We do not know if Smt3-3KR becomes more insoluble than wildtype SMT3. The blot in Figure 3D shows SUMO conjugates, not free SUMO so it is not possible to say if the free SMT3-3KR is more insoluble. In an attempt to measure this directly, we ran the whole cell, soluble and insoluble fractions for all samples (WT, Smt3-3KR +/- MMS), and probed for ubiquitin and SUMO. As noted in other studies (PMID: 22285753, PMID 23122649), DNA damage induces a wave of SUMOylation and ubiquitination and the same is evident in the whole cell and soluble fractions upon MMS treatment. Indeed, ubiquitin fractions were lower in whole cell extracts of Smt3-3KR MMS-treated cells indicative of the role of polySUMOylation in ubiquitinating proteins. Surprisingly, the insoluble pellet fractions did not contain any free SUMO or ubiquitin. Instead, only high molecular weight SUMOylated and ubiquitinated conjugates partitioned to the inclusion-containing pellet fractions. This also suggests that the Smt3 that localizes to INQ is conjugated to substrates and not present in a free SUMO form. Additionally, loss of polySUMOylation in Smt3-3KR along with retention of high molecular weight SUMO conjugates supports our notion of monoSUMOylation aiding in INQ formation while polySUMOylation might be involved in the removal of proteins from INQ.

4. In supplementary figure 3g, two closely migrating bands are observed in the SUMO pulldown, making it difficult to discern if Btn2-6KR is still highly SUMOylated or not, despite the arrow pointing to the band the authors appear to like the best. Can the authors comment on the identity of the extra band or perhaps obtain a cleaner result?

As noted in our response to Reviewer 1 we have now conducted the requested experiments. We have repeated GFP-Btn2 NiPD SUMO pulldowns in Btn2-WT/6KR backgrounds to attain a cleaner result and also in a *ubc9-1* background to test for the slower migrating band above Btn2 SUMO (see below).

Repeating this experiment in *ubc9-1* cells revealed a complete loss of Btn2 SUMOylation in Btn2-WT cells indicating that Btn2-6KR might still be SUMOylated at a different site (**Panel a**). However, quantification of Btn2 SUMOylation in both Btn2-WT and Btn2-6KR cells over multiple experiments revealed a 2.7-fold reduction of Btn2 SUMOylation in Btn2-6KR cells (**Panel b**). Upon closer inspection of **Figure 4b (Panel c, higher exposure)** and repeating experiments in untreated conditions, we do find that Btn2 is SUMOylated even in untreated conditions (**Panel d**). While this might suggest that Btn2 SUMOylation increases upon MMS treatment due to increase in protein levels, we see a significant reduction in Btn2 SUMOylation in Btn2-6KR cells, back to low basal levels without any change in Btn2 protein level under MMS indicating that we do in fact abolish the DNA damage induced SUMOylation in the Btn2-6KR mutant.

Therefore, while Btn2-6KR might still be SUMOylated at low basal levels, we believe DNA damage induced SUMOylation is significantly reduced in Btn2-6KR cells. We have reworded the paper to reflect the same.

5. The authors claim that the *Btn2-6KR* mutation does not affect solubility of *Btn2* protein *in silico* using protein structure prediction software. While these tools may suggest a likelihood, it would be far preferable if the authors could simply produce purified *Btn2* and *Btn2-6KR* from bacterial sources and validate this prediction. This is important, as the retention of the *KR* mutant protein in the pellet fraction (Fig. 5c) is either lack of SUMO or misfolding, two distinctly different rationales.

We were not able to produce purified *Btn2* (and the associated reagents to SUMOylate it) in the context of this paper but it will be an interesting and important future direction. Importantly, we did conduct filter trap assays to show that the insoluble material is indeed aggregated in a lysis-buffer resistant way. In terms of *6KR* mutation leading to misfolding of *Btn2* as a loss-of-function

we do not support this view because loss of Btn2 function through deletion of C-terminal truncation (**Figure 4f**) significantly reduces INQ formation. If the 6KR mutation inactivated Btn2 function as a sequestrase by misfolding it we would expect reduced INQ. Thus, Btn2-6KR acts as a functional sequestrase.

6. In Fig. 5d, the authors show that Btn2-6KR mutant highly increases Rpd3 foci. However, the effect of Btn2-6KR on other INQ markers is negligible, despite what seem to be aspirational significance numbers. For this reason, the claim in the title that SUMOylation of Btn2 “regulates nuclear protein quality control” seems an overreach. The authors are recommended to temper these claims with softer language, or identify additional substrates, even model ones, whose behavior mirrors that of Rpd3 more closely.

Thank you for the suggestion, we did not intend to overstate our results. It is important to note that in Figure 5D we did test the impacts of Btn2-6KR on multiple other endogenous INQ resident proteins and we see directional increases in foci levels for Hos2, Cmr1, Hsh155, Cdc48 and Slx8 although the effects were small in some cases. That said, we are happy to temper our language in the title.

We are a little unclear on the difference between regulator, modifier, or other descriptors. However, we have changed the title to “**DNA damage induced SUMOylation of Btn2 coordinates clearance of intranuclear inclusions**” for now. We are happy to work with the editor to find an appropriate title.

7. The result showing that Btn2-6KR partially ameliorates the cell growth defect of fes1 hsp104 mutant cells is extremely interesting and in truth, the sole point of physiological relevance for the SUMOylation discovery. However, this experiment doesn't link at all to the original point of the story regarding the DNA damage response. Can the authors add a panel with these strains and varying concentrations of MMS in the plates? Heightened resistance to the drug would be very compelling.

We thank the reviewer for this suggestion and agree that the partial rescue of the growth defect in *fes1Δhsp104Δ* cells is indeed very exciting. We repeated the spot assays under varying concentrations of MMS (see below) at 25°C to avoid complications introduced by heat stress. While there is a mild growth advantage in *fes1Δhsp104Δ*Btn2-6KR cells compared to *fes1Δhsp104Δ*Btn2-WT at 0.025% and 0.05% MMS concentrations, this phenotype is also visible in the untreated plate making it difficult to interpret the results. Since varying temperatures in **Figure 5f** has a clear, interpretable result, we have kept that as a part of the main figure and excluded the MMS experiment for now.

Other comments:

8. *The authors state in the methods that different statistical tests were used depending on the type of data – however, all figure legend indicate that only Fisher's test was used for every figure. Can the authors clarify?*

Except for the foci intensities compared in **Figure 5b** which used one-way ANOVA with multiple comparisons, the rest of the statistical tests were made using Fisher's exact test. We have corrected this mistake in the methods section.

9. *In contrast to the localization quantitation and statistical validation, no such rigor is found in the protein work – SUMOylation, subcellular fractionation, etc. Assuming the experiments were repeated multiple times, the authors should strongly consider quantitation of band intensities to strengthen their conclusions.*

We have added the quantification of Btn2 SUMOylation compared between Btn2-WT and Btn2-6KR and note a 2.7-fold reduction over multiple experiments. We also have quantified the fractionation data and included the relative fold changes in the respective figures. The raw data is included in the source file.

10. *The references for ubc9-1 ts mutant, DNA damaging agents, and Mms21-CH are missing.*

We have added the respective references.

11. *Fig. 6e mentioned in the text is Fig. 6f, vice versa.*

This mistake has been corrected.

12. *In Fig. S1C, the fractionation data of Ume1 is shown instead of Snt2 as mentioned in the text.*

This mistake has been corrected.

REVIEWER 3

In this study the authors investigated DNA damage-induced sequestration of Rpd3 (a histone deacetylase and a chromatin remodeler) to the intranuclear protein quality compartment (INQ), and the role of INQ-protein SUMOylation in this process, using yeast. Although Rpd3 localization to INQ has been previously identified in a large screen, this study investigates it in more detail. Following an earlier report of SUMO localization to the INQ, the authors show that mutants defective in protein SUMOylation have a defect in Rpd3 targeting to INQ. Next, the authors tested which INQ-proteins may be SUMOylated and found that a small chaperone Btn2 is SUMOylated upon DNA damage. To investigate the role of Btn2 SUMOylation in INQ, the

authors used a Btn2-6KR mutant. The data show that, upon removal of the DNA-damaging agent, cells expressing mutant Btn2-6KR have a delay in dissolution of Rpd3 from INQ, especially in the apj1 mutant background with impaired proteasomal degradation of INQ-components, indicating Btn2-SUMO function within the same epistasis group with Hsp104-dependent INQ-protein extraction.

Together the findings presented in the paper provide additional characterization of DNA damage induced Rpd3 sequestration to INQ, and raise an interesting possibility that DNA damage induced SUMOylation of a small chaperone is important for efficient protein dissolution from INQ upon stress removal.

I find the claims of DNA damage induced Rpd3 re-localization to INQ, in a process that is specific for Rpd3 subunit of different Rpd3-complexes and requires small chaperones Btn2 and Hsp42, well supported by the data, as well as DNA damage-induced SUMOylation of Btn2. The claims based on using Btn2-6KR mutant require additional clarification as pointed below.

Since the area of DNA replication stress is outside the scope of my expertise, I was unable to assess this part of the data fully.

The paper is written clearly.

Suggested improvements:

1. High molecular weight bands that correspond to SUMO-ylated Hsp42 in Fig. 4 b seem very weak compared to unmodified Hsp42, suggesting that only a very small fraction of Hsp42 is modified by SUMO. Could the authors provide a comment on this, modify the text accordingly or present additional data that corroborates substantial levels of Hsp42 SUMOylation.

It is true that only a small fraction of Hsp42 appears to be SUMOylated. We have not investigated SUMOylation of Hsp42 here, although it is certainly an area of interest. As reviewer 1 points out there have been reports of Hsp42 being SUMOylated in the literature, although our observations of DNA damage-induced SUMOylation is novel to our knowledge. Given the pleiotropic functions of Hsp42 we have not focussed on it in this work. As the reviewer suggests we have added a paragraph to the discussion to discuss the potential impact of small amounts of Hsp42 being SUMOylated.

2. In Western blot analyses of Ni-PD (Fig 4 g, Fig. S3-e) two bands are visible above non-modified Btn2, the lower one has been marked as corresponding to SUMO-ylated Btn2. The upper band is not visible in ctdD-mutant and empty vector control (Fig. 4 e), it is very faint in KallR mutant (Fig. S3-e, lane 3), but is stronger in the case of Btn2-6KR mutant (Figure 4 g lane 2, and Fig. S3-g lane 2).

Based on this, my concern is that Btn2-K6R mutant is still SUMOylated and, therefore, that the results using this mutant are difficult to interpret. Could the authors comment on what is the evidence for diminished Btn2-K6R mutant SUMOylation, i.e. what is the evidence that the upper

Kumar et al.

*band visible in Ni-PD of Btn2-K6R mutant is not a SUMOylated protein. For instance, does this band disappear in *ubc9ts* or SUMO-E3 ligase mutants?*

We thank the reviewer for this suggestion as it aligns with Reviewer 1. We have now conducted the requested experiments. We have repeated GFP-Btn2 NiPD SUMO pulldowns in Btn2-WT/6KR backgrounds to attain a cleaner result and also in a *ubc9-1* background to test for the slower migrating band above Btn2 SUMO (see below).

Repeating this experiment in *ubc9-1* cells revealed a complete loss of Btn2 SUMOylation in Btn2-WT cells indicating that Btn2-6KR might still be SUMOylated at a different site (**Panel a**). However, quantification of Btn2 SUMOylation in both Btn2-WT and Btn2-6KR cells over multiple experiments revealed a 2.7-fold reduction of Btn2 SUMOylation in Btn2-6KR cells (**Panel b**). Upon closer inspection of **Figure 4b (Panel c, higher exposure)** and repeating experiments in untreated conditions, we do find that Btn2 is SUMOylated even in untreated conditions (**Panel d**). While this might suggest that Btn2 SUMOylation increases upon MMS treatment due to increase in protein levels, we see a significant reduction in Btn2 SUMOylation in Btn2-6KR cells, back to low basal levels without any change in Btn2 protein level under MMS indicating that we do in fact abolish the DNA damage induced SUMOylation in the Btn2-6KR mutant.

Therefore, while Btn2-6KR might still be SUMOylated at low basal levels, we believe DNA damage induced SUMOylation is significantly reduced in Btn2-6KR cells. We have reworded the paper to reflect the same.

3. Following on the previous comment, Fig. 5 d shows that more cells expressing *Btn2-6KR* mutant display *Rpd3-GFP* INQ-foci, however, *SUMO-E3* mutants show a decrease in *Rpd3-GFP* INQ-foci (Fig. 3 b). Could the authors comment on this results.

Indeed, perturbation of global SUMOylation using the *SUMO-E2* and *-E3* mutants results in decreased *Rpd3* INQ formation. On the contrary, a *Btn2-6KR* mutant has increased

sequestration reminiscent of the polySUMOylation mutant Smt3-3KR. Our interpretation of these results was that global SUMOylation is important for INQ formation however SUMOylation of Btn2 is not. Based on washout experiments, Btn2 SUMOylation seems to be involved in INQ clearance instead. Since SUMO-E2 and -E3 mutants affect global SUMOylation, they quite possibly affect a multitude of processes governed by SUMOylation that regulate INQ formation. However, modification of Btn2 specifically seems to regulate the removal of proteins from INQ.

Minor comments:

4. In order to more easily understand the data showed in Fig. 2 b, I suggest moving the Figure S2 a and b to the main Fig. 2.

We thank the reviewer for this suggestion. **Supplementary Figure 2a** has now been moved to the main **Figure 2**.

*5. The result shown in Fig. 3a is difficult to interpret since percentage of cells with Rpd3-foci in *ubc9ts* mutant decreases in cells incubated both at 25 C and 30 C. Therefore the impact of the temperature itself interferes with the interpretation of the *ubc9ts* mutation. The data on the effect of impaired protein SUMOylation (E3 ligase mutants) is clear, so I suggest moving the figure with *ubc9ts* mutant to the Supplementary.*

We have conducted additional experiments at 25°C and show that the *ubc9-1* allele is hypomorphic at both 25°C and 30°C (new **Figure 4c**) as seen by the loss of Btn2 SUMOylation at both temperatures. We believe this will help interpret the results of **Figure 3a** and that the reduction of Rpd3 INQ foci at both temperatures is due to perturbation of SUMOylation.

6. Paragraph lines 183-193: has the effect of SUMOylation mutants been tested for Btn2-GFP and/or other INQ-components? If this conclusion does not refer to INQ in general, I recommend that authors specify that the conclusion refers to Rpd3-INQ.

We have tested the effect of some SUMOylation mutants in our previous study with another INQ protein Hsh155 (PMID 28978642) which served as preliminary results and formed the basis of this study. However, we agree with the reviewer that the results listed in this manuscript are specific to Rpd3 and have amended the text to reflect this.

REVIEWER COMMENTS

Reviewer #1 (Remarks to the Author):

The revised manuscript is much improved. The authors have performed several additional experiments, including using Nic96-RFP as a marker of the nuclear boundary and the filter trap assay in Fig 5c. These new data and further clarifications in the text have addressed my concerns.

Very minor comments:

Line 337 should refer to Fig 5d instead of 5e.

The CHX chase experiment in Supplemental figure 5c shows the PGK loading control also decreasing over time. Given that the stabilization of GFPBtn2 in that blot is minimal, the conclusion that Slx5/8 could be involved in the turnover of Btn2 in non-stressed conditions is a minor point, and the contribution of Slx5/8 to the SUMOylation of Btn2 and INQ is convincingly shown in the main figure, I'm not sure that this data is adding anything substantial to the manuscript. A blot that clearly demonstrates stabilization would be more convincing.

Reviewer #2 (Remarks to the Author):

The authors have nearly addressed all of my significant concerns, several of which were also noted by the other reviewers. Additional experiments and better quantitation of protein work (gels/blots) are welcomed and appropriate.

I and another reviewer had questions regarding potential differential solubility of the Btn2-6KR mutant protein that were addressed by the authors using a filter trap assay. While this is the appropriate experiment, the authors must add load controls for the total cell extract applied to the filter to discern whether the band intensities for the two samples reflect differential retention or differential protein levels in the original extract (Fig. 5C). This new experiment is inconclusive and incomplete otherwise.

Reviewer #3 (Remarks to the Author):

Here are my comments on the revised version of the manuscript:

1. As the authors discuss in the Rebuttal letter, GFP-Btn2 seems to be SUMOylated even in the absence of DNA damage - see Fig. 4 b (compare lanes 3 and 4 taking into account input levels), and Fig. Panel d from the Rebuttal letter (compare lanes 1 and 2 taking into account input levels). GFP-Btn2-6KR seems to give the same pattern with or without MMS-treatment (figure Panel d from Rebuttal letter, compare lanes 3 and 4).

Based on the data mentioned above, I am afraid I cannot see the evidence for "DNA damage induced SUMOylation of Btn2" (as is written in the new title).

What I do find the evidence for is:

- DNA damage induced relocalization of Rpd3 into nuclear inclusions (INQ), which is dependent on Btn2 (Fig. 1, Fig. 2) and general SUMOylation (ubc9ts, E3 - Fig. 3),
- disassembly/ clearance of Rpd3-INQ is delayed in Btn2-6KR mutant (Fig. 6 A)
- Btn2-6KR mutant appears less SUMOylated than the wild-type GFP-Btn2 (Fig. 4g; Fig S3g; very weak difference in Panel d from Rebuttal letter)
- About the evidence that Btn2 is modified by SUMO: As the authors discuss in the Rebuttal letter, GFP-Btn2 seems to stick to the Ni-beads even in the absence of Hisx7-Smt3 expression (Fig. 4 b, compare lanes 2 and 4), making it uncertain whether the slower-migrating band is really SUMO-

modified GFP-Btn2, rather than another posttranslational modification. The only evidence that this upper band is a SUMO-modification is that it disappears in the *ubc9ts* mutant (Fig. 4c).

Based on above, I suggest not using the phrase "DNA damage induced SUMOylation of Btn2" in the title and changing it into something like: "DNA damage-induced (sequestration of proteins/ Rpd3 into) nuclear inclusions are regulated by SUMOylation (of Btn2)", or "Dynamics of DNA damage-induced nuclear inclusions is regulated by SUMOylation (of Btn2)".

I suggest modifying the phrasing like "DNA damage induced SUMOylation of Btn2" in the Abstract (the current text is: "...SUMO modification of the small heat shock protein Btn2 occurs in response to DNA damage") and throughout the manuscript (line 225, the title of the sub-chapter is "DNA damage induces SUMOylation of sequestrase chaperones")

2. I could not find Panel d from Rebuttal letter in the revised manuscript.

differential protein levels vs different retention on the membrane. To this extent, we ran an SDS-PAGE to showcase no significant change in GFPBtn2 protein levels, thus attributing the increased insolubility to loss of SUMOylation in the mutant (similar to PMID 37118550 and 35880124). We have now updated the figure legend and quantified the band intensities to further support the data.

REVIEWER 3:

Here are my comments on the revised version of the manuscript:

1. As the authors discuss in the Rebuttal letter, GFP-Btn2 seems to be SUMOylated even in the absence of DNA damage - see Fig. 4 b (compare lanes 3 and 4 taking into account input levels), and Fig. Panel d from the Rebuttal letter (compare lanes 1 and 2 taking into account input levels). GFP-Btn2-6KR seems to give the same pattern with or without MMS-treatment (figure Panel d from Rebuttal letter, compare lanes 3 and 4).

Based on the data mentioned above, I am afraid I cannot see the evidence for “DNA damage induced SUMOylation of Btn2” (as is written in the new title).

What I do find the evidence for is:

- DNA damage induced relocalization of Rpd3 into nuclear inclusions (INQ), which is dependent on Btn2 (Fig. 1, Fig. 2) and general SUMOylation (ubc9ts, E3 – Fig. 3),
- disassembly/ clearance of Rpd3-INQ is delayed in Btn2-6KR mutant (Fig. 6 A)
- Btn2-6KR mutant appears less SUMOylated than the wild-type GFP-Btn2 (Fig. 4g; Fig S3g; very weak difference in Panel d from Rebuttal letter)
- About the evidence that Btn2 is modified by SUMO: As the authors discuss in the Rebuttal letter, GFP-Btn2 seems to stick to the Ni-beads even in the absence of Hisx7-Smt3 expression (Fig. 4 b, compare lanes 2 and 4), making it uncertain whether the slower-migrating band is really SUMO-modified GFP-Btn2, rather than another posttranslational modification. The only evidence that this upper band is a SUMO-modification is that it disappears in the ubc9ts mutant (Fig. 4c).

Based on above, I suggest not using the phrase “DNA damage induced SUMOylation of Btn2” in the title and changing it into something like: “DNA damage-induced (sequestration of proteins/ Rpd3 into) nuclear inclusions are regulated by SUMOylation (of Btn2)”, or “Dynamics of DNA damage-induced nuclear inclusions is regulated by SUMOylation (of Btn2)”.

I suggest modifying the phrasing like “DNA damage induced SUMOylation of Btn2” in the Abstract (the current text is: “...SUMO modification of the small heat shock protein Btn2 occurs in response to DNA damage”) and throughout the manuscript (line 225, the title of the sub-chapter is “DNA damage induces SUMOylation of sequestrase chaperones”

Indeed, Btn2 appears to be SUMOylated in the absence of DNA damage. Upon MMS treatment, we see an increase in Btn2 protein levels and also SUMOylation, suggesting that the increased SUMOylation could simply be due to more protein. However, we believe DNA damage does in fact induce SUMOylation of Btn2. Note that in Btn2-6KR cells treated with MMS SUMOylation is significantly reduced to what looks like untreated Btn2 SUMOylation levels. Importantly, in this situation, Btn2 protein levels are still induced by MMS, despite seeing less SUMO. Nevertheless, we appreciate the reviewer's concern about the language and have amended the title and text to match their perspective. The new title is: "Dynamics of DNA damage-induced nuclear inclusions are regulated by SUMOylation of Btn2"

Text changes were made from "DNA damage induces/induced SUMOylation" to "SUMOylation under DNA damage":

Line 17: "SUMO modification of the small heat shock protein Btn2 under DNA damage"

Line 92: "we observe SUMOylation of both sHsps Btn2 and Hsp42 under DNA damage"

Line 229: "On the contrary, we noted clear SUMOylation of both sHsp sequestrases Btn2 and Hsp42 but for none of the other chaperones tested under DNA damage"

Line 280: "In summary, we reveal SUMOylation of the sHsps Btn2 and Hsp42 under DNA damage."

2. I could not find Panel d from Rebuttal letter in the revised manuscript.

This rebuttal figure was intended to clarify a reviewer's concern with replicates showing the basal levels of Btn2-SUMO. This data is also evident in the main text Figure 4b (3rd lane) which we mentioned in Line 274:

"It is important to note that we observe a weaker DNA-damage insensitive SUMOylation in Btn2-6KR (quantified in **Supplementary Fig. 3g**). We know this is likely SUMO because it was abolished in *ubc9-1* alleles, and suggests that Btn2 may be SUMOylated constitutively at some level on other sites (**Fig. 4c**). Indeed, we also note a low level of SUMOylation for Btn2 in untreated cells (**Fig. 4b**). Importantly, Btn-6KR reduces DNA damage induced SUMO by almost 3-fold (**Supplementary Fig. 3g**) without affecting Btn2 protein levels under MMS, making it a useful tool to probe the impact of this PTM."

We felt that the additional blot we ran for the rebuttal was redundant and did not include it in the revised manuscript. We are happy to add it to a supplementary Figure if you would like.